# Task-Aware Exploration via a Predictive Bisimulation Metric

**Dayang Liang**[1]   **Ruihan Liu**[1 2]   **Lipeng Wan**[3]   **Yunlong Liu**[⋆ 1]   **Bo An**[4]

## Abstract

Accelerating exploration in visual reinforcement learning under sparse rewards remains challenging due to the substantial task-irrelevant variations. Despite advances in intrinsic exploration, many methods either assume access to low-dimensional states or lack task-aware exploration strategies, thereby rendering them fragile in visual domains. To bridge this gap, we present **TEB**, a **T**ask-aware **E**xploration approach that tightly couples task-relevant representations with exploration through a predictive **B**isimulation metric. Specifically, TEB leverages the metric not only to learn behaviorally grounded task representations but also to measure global intrinsic novelty over the learned latent space. To realize this, we first theoretically mitigate the representation collapse of degenerate bisimulation metrics under sparse rewards by internally introducing a simple but effective predicted reward differential term. Building on this robust metric, we further introduce potential-based global exploration bonuses over anchor states, which measure the relative novelty between observations over the latent space. Extensive experiments on MetaWorld and Maze2D show that TEB achieves superior exploration ability and outperforms recent baselines. Code is available at https://github.com/dy-liang/teb.

## 1. Introduction

Visual reinforcement learning (RL) has made significant progress in continuous control (Chen et al., 2024; Mu et al.,

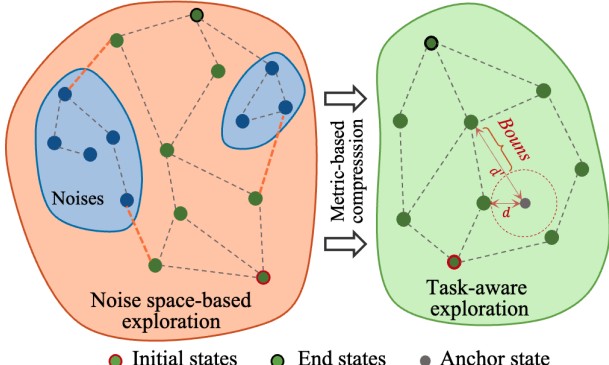

*Figure 1.* An illustration of existing exploration strategies in noise space and our task-aware exploration in clean space. In complex tasks, existing strategies remain limited by task-irrelevant elements, resulting in risky transitions like the "orange line". In contrast, the task-relevant space and exploration bonus built on our bisimulation metric can incentivize visual RL to complete tasks faster. The right figure illustrates how to construct a global exploration bonus that measures relative novelty between states from a global anchor state using the same metric in a rigorous task-relevant space.

2025; Yang et al., 2025; Wang et al., 2025), highlighting the potential to learn decision policies directly from high-dimensional observations. However, in complex domains, particularly those with sparse or delayed rewards, it remains challenging to improve extrinsic-reward exploration and sample efficiency. To address this, prior work has developed intrinsic rewards based on novelty or uncertainty, including forward prediction-error curiosity (Pathak et al., 2017), Bayesian Surprise (Mazzaglia et al., 2022), random network distillation (Burda et al., 2018), and ensemble disagreement, as well as density-based objectives such as kNN entropy estimation (Mutti et al., 2021) and state-marginal matching (Lee et al., 2019). Nonetheless, these intrinsic rewards are often defined over latent spaces that are not rigorously aligned with task semantics (Pathak et al., 2017; Yarats et al., 2021b). They are therefore susceptible to nuisance variations such as texture, illumination, and background noise, which can steer behavior away from the core task (Bu et al., 2024). In addition, unsupervised skill discovery methods (Yang et al., 2023; Bai et al., 2024; Sun et al., 2025) expand coverage by maximizing skill discriminability or temporal reachability. Although they can provide coverage benefits in theory, they typically rely on accessible ground-truth states and incur substantial overheads, which limits their efficiency

---

⋆Corresponding author [1]Department of Automation, Xiamen University, Xiamen, China [2]Department of Computer Science, Beijing Normal-Hong Kong Baptist University, Zhuhai, China [3]School of Artificial Intelligence, Xi'an Jiaotong University, Xi'an, China [4]College of Computing and Data Science, Nanyang Technological University, Singapore. Correspondence to: Yunlong Liu <ylliu@xmu.edu.cn>.

*Proceedings of the 43rd International Conference on Machine Learning*, Seoul, South Korea. PMLR 306, 2026. Copyright 2026 by the author(s).

and scalability in practical visual environments.

This motivates a central question: **can we develop a simple mechanism that both captures task structure and quantifies task-relevant exploration bonuses?** Bisimulation metrics provide a principled foundation for this goal. By combining differences in rewards and dynamic transitions, bisimulation and its variants measure behavioral equivalence between states (Ferns et al., 2004; 2011), which have previously been used to learn mappings from high-dimensional observations to task-relevant latent spaces (Zhang et al., 2021; Zang et al., 2024). In fact, the metric itself offers a task-aware notion of distance (Liu et al., 2023), making it a natural candidate for constructing exploration bonuses. Moreover, existing theory shows that the state value difference is bounded by the bisimulation distance (Chen & Pan, 2022), which suggests that larger bisimulation distances may correspond to larger disparities in long-term returns. These insights motivate a unified perspective in which the same bisimulation metric can underpin both representation learning and intrinsic-reward construction. However, despite attempts made by existing work (Wang et al., 2023), they have struggled to achieve task-aware exploration with fragile metric representations. This is primarily because the metric collapse problem remains unresolved, i.e., the bisimulation metrics are still vulnerable to sparse or delayed rewards leading to metric degradation and representation collapse (Chen & Pan, 2022; Liao et al., 2023). In addition, naively leveraging bisimulation metrics to calculate exploration bonuses can easily get stuck in local exploration.

To address these issues, we propose **TEB**, a task-aware exploration approach built on a predictive bisimulation metric. The core idea is to keep the bisimulation metric non-degenerate and representations learnable under sparse rewards, while leveraging the same metric principle to shape global exploration bonuses. We briefly summarize the insights in Figure 1. Concretely, we first introduce a reward predictor that produces Gaussian reward differentials to replace the extrinsic reward differentials used in conventional bisimulation metrics. This prediction-based reward can theoretically guarantee a positive metric radius between states even in the worst-case transitions, which mitigates representation collapse caused by zero-distance metrics. Building on this robust metric, we construct task-aware intrinsic exploration bonuses. Specifically, we introduce a potential function (Ng et al., 1999; Wang et al., 2023) based on the proposed metric and then compute global novelty through relative potential differences from an approximately global anchor state. This design effectively encourages task-aware exploration toward global regions while preserving the optimal policy in theory.

We extensively evaluate TEB on MetaWorld (Yu et al., 2020) and Maze2D (Campos et al., 2020) environments. The re-

sults demonstrate that TEB achieves superior policy performance in challenging visual tasks and also outperforms powerful intrinsic exploration baselines on low-dimensional Maze2D tasks. Ablation studies confirm the effectiveness of TEB components. Our contributions are threefold.

(1) We propose a visual RL method that tightly couples task representations and task-aware exploration through a predictive bisimulation metric.

(2) We propose a simple yet effective predictive bisimulation metric that extends the metric-based representation to sparse-reward visual settings.

(3) We introduce a metric-based intrinsic reward that encourages agents to perform global task-aware exploration in the representation space.

## 2. Preliminaries

We assume that the underlying environment is a discounted Markov decision process (MDP) defined by the tuple $\mathcal{M} = \langle \mathcal{S}, \mathcal{A}, \mathcal{P}, r, \gamma \rangle$, where $\mathcal{S}$ denotes the state space, $\mathcal{A}$ the action space, $\mathcal{P}(s'|s, a)$ the state transition function that captures the probability from $s$ to $s'$ given action $a$, $r(s, a) \in \mathbb{R}$ the reward function, and $\gamma \in [0, 1)$ the discount factor. A policy $\pi(a|s)$ denotes the probability of selecting action $a$ conditioned on state $s$. The value function $V^\pi : \mathcal{S} \to \mathbb{R}$ is defined as the expected cumulative discounted rewards,

$$V^\pi(s) = \mathbb{E}_{\substack{a_t \sim \pi \\ s_{t+1} \sim \mathcal{P}}} \left[ \sum_{t=0}^{\infty} \gamma^t r(s_t, a_t) | s_0 = s \right] \quad (1)$$

The RL goal is to learn an optimal policy $\pi^* = \operatorname{argmax}_\pi V^\pi$ that maximizes cumulative discounted rewards. In high-dimensional vision tasks, we also jointly learn an encoder $\phi_\omega : \mathcal{S} \to \mathcal{Z}$ that maps high-dimensional observations $s \in \mathcal{S}$ to a low-dimensional latent space $z \in \mathcal{Z}$, while retaining only task-relevant information for learning a parameterized policy $\pi(a|\phi_\omega(s))$.

### 2.1. Bisimulation Metrics

The bisimulation metric (Ferns et al., 2004; 2011) measures behavioral similarity between states by defining a pseudo-metric $d : \mathcal{S} \times \mathcal{S} \to \mathbb{R}_{\geq 0}$ over a reward differential term and a 1-Wasserstein distance in dynamics models. Theorem 2.1 is a policy-dependent bisimulation metric, which focuses on behaviors relative to a specific policy.

**Theorem 2.1** ($\pi$-bisimulation metric (Castro, 2020) )**.** *Given a MDP $\mathcal{M}$ and a fixed policy $\pi$, the following on-policy bisimulation metric exists and is unique:*

$$\begin{aligned} d^\pi(s_i, s_j) &= c_R \left| r_{s_i}^\pi - r_{s_j}^\pi \right| \\ &+ c_T \mathcal{W}_1(d^\pi)(\mathcal{P}^\pi(\cdot|s_i), \mathcal{P}^\pi(\cdot|s_j)), \end{aligned} \quad (2)$$

*where the $r_s^\pi = \mathbb{E}_{a \sim \pi}[r(s, a)]$, $\mathcal{P}_{s_i}^\pi = \mathbb{E}_{a \sim \pi} \mathcal{P}^\pi(\cdot|s_i)$ and*

$\mathcal{W}_1$ *is the 1-Wasserstein distance.* $c_R > 0$ *and* $c_T \in [0, 1)$ *are the coefficients of the reward and transition terms of the metric, respectively.* $d^\pi(s_i, s_j)$ *has a least fixed point* $d_B^\pi$ *and* $d_B^\pi$ *is a* $\pi$*-bisimulation metric.*

The central idea behind bisimulation metric-based representation learning is to embed task-relevant features that are associated with the abstract semantics of both rewards and transition dynamics, from high-dimensional observations into a compact latent space (Zhang et al., 2021). This is typically achieved by minimizing the discrepancy between the latent distance $\|\phi_\omega(s_i) - \phi_\omega(s_j)\|$ and the bisimulation distance $d^\pi(s_i, s_j)$ for any state pairs $s_i$ and $s_j$,

$$\mathcal{L}_{\text{bisim}} = \mathbb{E}\left[\left(\|\phi_\omega(s_i) - \phi_\omega(s_j)\| - d^\pi(s_i, s_j)\right)^2\right] \quad (3)$$

# 3. Method

In this section, we present task-aware exploration with a predictive bisimulation metric (TEB) in detail. The proposed method aims to tightly couple visual representation learning with intrinsic exploration by constructing a robust bisimulation metric. We begin by analyzing the limitations of conventional bisimulation metric-based representation learning under sparse rewards. We then introduce a predictive bisimulation metric that remains non-degenerate under sparse rewards and provide theoretical guarantees. Finally, we design a potential-based exploration bonus derived from this metric over the learned task-relevant space and prove its invariance with respect to the optimal policy.

## 3.1. The Risk of Bisimulation Metrics

Bisimulation metrics for learning a grounded task-relevant representation space hold great promise. To make them scalable, prior work introduces the policy-dependent bisimulation metric $d^\pi(s_i, s_j)$, which estimates behavioral similarity from transition samples collected under a fixed policy (Castro, 2020), making it practical and tractable. Despite this appeal, the metric remains challenging to scale in sparse reward settings. Intuitively, when $|r_{s_i}^\pi - r_{s_j}^\pi|$ is frequently near zero, the fixed-point recursion in Eq. (2) aggressively contracts distances as the reward term provides negligible scale, and the Wasserstein term is discounted by $c_T < 1$. Theoretically, existing analysis shows that the metric diameter is upper bounded by reward differentials.

**Lemma 3.1** (Diameter of $\mathcal{S}$ is bounded (Kemertas & Aumentado-Armstrong, 2021)). *On-policy* $\pi$*-bisimulation metric* $d^\pi : \mathcal{S} \times \mathcal{S} \to [0, \infty)$ *has a fixed upper bound on the diameter of* $\mathcal{S}$.

$$\text{diam}(\mathcal{S}; d^\pi) \leq \frac{c_R}{1 - c_T} \max_{s_i, s_j} |r_{s_i}^\pi - r_{s_j}^\pi|. \quad (4)$$

Lemma 3.1 suggests that the bisimulation metric is theoretically dependent on the inner reward differential term.

If the immediate reward signal remains within a negligible range over a long period, for instance, $r_s^\pi \equiv 0$, then the bisimulation metric degenerates to the trivial solution $\text{diam}(\mathcal{S}, d^\pi) = 0$, which in turn induces an erroneous collapse of task-relevant distances in the latent representation.

## 3.2. The Proposed Predictive Bisimulation Metric

**Predictive Bisimulation Metric.** To prevent representation collapse caused by zero-distance metric under sparse reward settings, we propose the robust predictive bisimulation metric by replacing the reward differential with a predictive reward differential $\Delta_R^\pi(s_i, s_j)$. Our predictive bisimulation metric operator on ground-truth states is defined by

$$\begin{aligned}(\widehat{\mathcal{T}}^\pi d_{\text{pre}})(s_i, s_j) &= c_R \, \Delta_R^\pi(s_i, s_j) \\ &+ c_T \, \mathcal{W}_1(d_{\text{pre}})\left(\widehat{\mathcal{P}}^\pi(\cdot|s_i), \widehat{\mathcal{P}}^\pi(\cdot|s_j)\right)\end{aligned} \quad (5)$$

with

$$\Delta_R^\pi(s_i, s_j) = \mathbb{E}_{a_i, a_j \sim \pi}\left[\left|\hat{r}_\theta(s_i, a_i) - \hat{r}_\theta(s_j, a_j)\right|\right], \quad (6)$$

where $c_R > 0$ and $c_T \in [0, 1)$. $\hat{r}_\theta(s, a)$ denotes a random variable sampled from the predicted reward distribution $p_\theta(\cdot|s, a)$ with parameters $\theta$ under the action induced by a specific policy. $\widehat{\mathcal{P}}^\pi = \mathbb{E}_{a \sim \pi}\widehat{\mathcal{P}}(s_i, a)$ is a learned policy-marginal transition model. When $\hat{r}_\theta(s, a) = r(s, a)$ and $\widehat{\mathcal{P}}^\pi = \mathcal{P}^\pi$, Eq. (5) reduces to the standard on-policy bisimulation metric.

**Lemma 3.2** (Convergence and Fixed Point). *Predictive bisimulation metric* $\widetilde{d}_{\text{pre}}$ *is a contraction mapping w.r.t the* $L^\infty$ *norm on* $\mathbb{R}^{\mathcal{S} \times \mathcal{S}}$, *and there exists a fixed-point* $\widetilde{d}_{\text{pre}}$.

The proof of Lemma 3.2 is available in Appendix A.1. This provides a theoretical convergence property for bootstrapped metric and representation learning based on the predictive bisimulation metric.

**Gaussian Reward Predictor.** We assume that the external reward is Gaussian distributed. Then, let $p_\theta(\cdot|s_t, a_t)$ be a Gaussian predictor, which outputs a Gaussian distribution conditioned on the current state and action. We model it on the latent space $\mathcal{Z}$ by,

$$p_\theta(r_t|s_t, a_t; \omega) = \mathcal{N}\left(\mu_\theta(z_t, a_t), \sigma_\theta^2(z_t, a_t)\right) \quad (7)$$

where the mean $\mu_\theta$ and the variance $\sigma_\theta$ are learned networks, as well as the latent state $z_t = \phi_\omega(s_t)$. We train the reward predictor by minimizing the negative log-likelihood on replay data,

$$\begin{aligned}\mathcal{L}_{\text{rew}}(\theta) &= -\mathbb{E}_{\mathcal{D}}\left[\log p_\theta(r_t|s_t, a_t; \omega)\right] \\ &= \mathbb{E}_{\mathcal{D}}\left[\frac{(r_t - \mu_\theta(z_t, a_t))^2}{2\sigma_\theta(z_t, a_t)^2} + \frac{1}{2}\log \sigma_\theta(z_t, a_t)^2\right].\end{aligned} \quad (8)$$

In the training process, the target rewards are the multi-step Monte Carlo return, i.e., $r_t = \sum_{k=0}^{N-1} \gamma^k r_{t+k}$, and the variance $\sigma_\theta$ is scaled to $[\sigma_{\min}, \sigma_{\max}]$, where $\sigma_{\min}$, $\sigma_{\max}$ and $N$ are detailed in the Appendix D.2.

The learned predictor $p_\theta(\cdot|s_t, a_t; \omega)$ leverages $\mu_\theta$ to estimate the conditional expectation of the target $r_t$, and leverages $\sigma_\theta$ to characterize the scale of the stochastic residual under the given state and policy. By reparameterization, a predictive sample can be written as $\hat{r}_\theta(s_t, a_t) = \mu_\theta + \sigma_\theta \odot \xi$ with $\xi \sim \mathcal{N}(0, I)$, and hence admits the decomposition $\hat{r}_\theta(s_t, a_t) = r_t + \varepsilon_t$ where $\varepsilon_t$ collects the remaining approximation error $\mu_\theta - r_t$ and the stochastic component induced by $\sigma_\theta$. Due to the optimized tradeoff of the variance term in the denominator (in $\mathcal{L}_{\text{rew}}$) and the enforced lower bound $\sigma_{\min}$, it implies a positive variance w.r.t $\sigma_\theta$, especially relevant in early learning, which thereby supports the variance assumption required by Theorem 3.3.

**Non-degenerate Theoretical Analysis.** Next, we leverage Theorem 3.3 to show that the bisimulation metric with the predictive reward yields a non-degenerate distance, thereby offering theoretical justification for learning a robust representation space under sparse reward conditions.

**Theorem 3.3** (Predictive Reward Prevents Degenerate Metric)**.** *Let the state space $\mathcal{S}$ be compact and the policy $\pi$ fixed. Consider the predictive bisimulation operator $\widehat{\mathcal{T}}^\pi$, the learned reward model satisfies $\hat{r}_\theta(s_t, a_t) = r_t + \varepsilon_t$ with $\mathrm{Var}[\varepsilon_t | s_t, a_t] = \Sigma > 0$. Then, the unique fixed point $\widetilde{d}_{pre} = \widehat{\mathcal{T}}^\pi \widetilde{d}_{pre}$ admits a strictly positive expected diameter in the sparse-reward region $\mathcal{S}_0 = \{s | r(s,a) = 0\}$:*

$$\mathbb{E}_{\mathcal{D}}\left[\mathrm{diam}(\mathcal{S}_0; \widetilde{d}_{pre})\right] \in \left(0, \frac{c_R}{1 - c_T} \cdot \frac{2}{\sqrt{\pi}}\sqrt{\Sigma}\right]. \quad (9)$$

The proof of Lemma 3.3 is available in Appendix A.2. This result formally shows that the predictive bisimulation employed in TEB preserves a nonzero geometric scale within the latent space. While the classical $\pi$-bisimulation metric collapses in sparse-reward environments because the reward differential $|r_{s_i}^\pi - r_{s_j}^\pi|$ vanishes, TEB's predictive reward model introduces a stochastic residual $\varepsilon_t$ with finite variance $\Sigma$. This residual induces a positive expected separation between latent states, thereby preventing the bisimulation metric from degenerating into a zero distance. Intuitively, the predictor's uncertainty injects a minimal "energy floor" into the metric-based representation learning, which ensures that the encoder $\phi_\omega$ continues to receive meaningful gradient signals and that the learned latent manifold remains well-structured even when external rewards are sparse or absent. In practice, this mechanism directly explains TEB's empirical stability and strong generalization under sparse reward settings.

Besides this, another situation also empirically favors the non-degenerate advantage of our predictive bisimulation

metrics. Specifically, when individual rewards occur, the reward predictor can also leverage the network's inductive and generalization abilities to ensure that many similar transitions generate positive rewards, thus enabling the sampled $\Delta_R^\pi(s_i, s_j) > 0$. This advantage mainly occurs in the early stages of training because the predictor has not yet overfitted. Nonetheless, the above is an empirical advantage we speculate on and do not focus on.

**Training Loss.** Let $\hat{d}_\omega(s_i, s_j)$ be the approximated bisimulation metric distance $\|\phi_\omega(s_i) - \phi_\omega(s_j)\|$ between $s_i$ and $s_j$ parameterized by $\omega$. Finally, we train the encoder by a bootstrapped bisimulation regression loss:

$$\mathcal{L}_{\text{bisim}}(\omega) = \mathbb{E}_{s_i, s_j \sim \mathcal{D}}\left[\hat{d}_\omega(s_i, s_j) - \mathrm{sg}[\widehat{\mathcal{T}}^\pi d_{\text{pre}}](s_i, s_j; \bar{\omega}))^2\right], \quad (10)$$

where $\mathrm{sg}[\cdot]$ denotes stop gradients with the frozen parameters, e.g. $\bar{\omega}$ and $\bar{\theta}$. In addition, we need to learn a dynamic transition model through common loss structure $\mathcal{L}_{\text{dyn}}$ following Chen et al. (Chen & Pan, 2022). The overall representation objective is $\mathcal{L}_{\text{rep}} = \lambda_b \mathcal{L}_{\text{bisim}} + \lambda_r \mathcal{L}_{\text{rew}} + \lambda_p \mathcal{L}_{\text{dyn}}$, where the coefficients are referenced from previous work (Chen & Pan, 2022) and detailed in the Appendix D.2.

### 3.3. Metric-based Intrinsic Exploration

**Link to Intrinsic Exploration.** The definition of the bisimulation metric directly indicates that it can measure behavioral similarity by constructing a distance based on the difference in rewards and the dynamic transition distribution between two states. This inspires us to utilize the property directly to measure the behavioral distance relevant to the task in the latent space $\mathcal{Z}$, and then to carefully design an intrinsic bonus based on this distance that can facilitate the exploration of reinforcement learning. Furthermore, an important characteristic of the connection between the bisimulation metrics and the value function further suggests the above insight.

**Theorem 3.4** (Value Difference Bound)**.** *Given states $s_i$ and $s_j$, and a policy $\pi$, let $V(s_t)$ be the state value function over policy $\pi$, we have,*

$$|V^\pi(s_i) - V^\pi(s_j)| \le \widetilde{d}_{pre}(s_i, s_j) \quad (11)$$

The proof of Theorem 3.4 is available in Appendix A.3. It shows that given any two states, the difference in their value functions is bounded by the predictive bisimulation metric. In addition, the theory suggests that distant states in the metric possibly differ substantially in long-term value, while simultaneously increasing the probability of discovering high-value states. The feature also motivates us to construct meaningful exploration bonuses by the bisimulation metric.

**Metric-based Potential Function.** Nevertheless, directly using this metric $d_{\text{pre}}(s_i, s_j)$ itself as an intrinsic reward

does not reliably reflect meaningful novelty, and it may encourage agents to make local transitions. Instead, following Wang et al. (Wang et al., 2023), we build a relative novelty signal via a potential function defined in the learned latent space. Distinctly, we start with the aforementioned value differential interpretation perspective, and improve upon the shortcomings of existing work in local exploration by designing global exploration with global anchor states.

First, we define a metric-based potential with pseudo-anchor state $s_\star$, which aims to potentially measure the absolute task-relevant novelty of the current state in a batch of samples with a batch size of $B$,

$$
\begin{aligned}
\Phi(s_t) &= d_{\text{pre}}(s_t, s_\star; \bar{\omega}) \\
&= c_R \big| \hat{r}_\theta(s_t, a_t; \bar{\omega}) - \hat{r}_\star \big| + c_T \, \mathcal{W}_1\big(\widehat{\mathcal{P}}(s_t, a_t; \bar{\omega}), z_\star'\big)
\end{aligned}
\tag{12}
$$

with

$$
r_\star = \frac{1}{B} \sum_{k=1}^{B} \mathbf{r}_{(k)}, \; z_\star' = \frac{1}{B} \sum_{k=1}^{B} \mathbf{z}_{(k)}'
\tag{13}
$$

where $r_\star$ and $z_\star'$ denote the anchor reward and the latent anchor state averaged from batch transitions in the training stage, as well as $\mathbf{z}_{(k)}' = \widehat{\mathcal{P}}(s_k, a_k; \bar{\omega})$. Particularly, $z_\star'$ calculates the element-wise average of a batch of next latent states across hidden neurons. Therefore, we employ this abstract average information as anchors to approximate the global novelty of the current latent state.

Then, we introduce potential-based shaping to compute the intrinsic exploration bonus between adjacent latent states. Specifically, it calculates the relative discounted distance between consecutive metric-based potentials as the intrinsic bonus, i.e., $F(s_t, a, s_{t+1})$, thus producing a meaningful exploration bonus that approximates the novelty relevant to the global task.

$$
\begin{aligned}
F(s_t, a, s_{t+1}) &= \gamma \, \Phi(s_{t+1}) - \Phi(s_t) \\
&= \gamma \, d_{\text{pre}}(s_{t+1}, s_\star; \bar{\omega}) - d_{\text{pre}}(s_t, s_\star; \bar{\omega}),
\end{aligned}
\tag{14}
$$

Finally, we define the intrinsic reward $r_{int} = F(s_t, a, s_{t+1})$ and a new reward function that aggregates the external reward and the intrinsic reward, i.e., $r' = r + \eta \, r^{\text{int}}$, to train the temporal difference loss. The selection of $\eta$ can be found in Appendix D.2.

Although the potential function $\Phi(s)$ is constructed using both the current state and the pseudo-anchor state, the policy invariance in our setting can still be analyzed through the standard potential-based shaping argument in an update-wise manner. Specifically, once a minibatch is sampled for the current policy optimization step, the anchor $s_\star$ is fixed within that update. Therefore, during the current update period, the potential in Eq. (12) depends only on the state and can be viewed as a standard potential function, so that the shaping bonus in Eq. (14) admits the usual policy-invariance

interpretation. When the anchor is recomputed across updates, the resulting practical variant becomes non-stationary and can be further understood through the lens of dynamic potential functions (Devlin & Kudenko, 2012). In this sense, we still use the standard potential-based reward shaping argument to informally justify the policy invariance of TEB in the batch-based anchor state setting.

**Theorem 3.5** (Policy Invariance of Metric-based Shaping). *Let $\mathcal{M} = (\mathcal{S}, \mathcal{A}, \mathcal{P}, r, \gamma)$ denote the original MDP, and consider the modified reward function $r' = r + \eta \, F(s, a, a')$, where $F$ is defined in Eq (14). Then the modified MDP $\mathcal{M}' = (\mathcal{S}, \mathcal{A}, \mathcal{P}, r', \gamma)$ with the metric-based potential preserves the optimal policy of $\mathcal{M}$ by,*

$$
Q_{\mathcal{M}'}^*(s, a) = Q_{\mathcal{M}}^*(s, a) - \eta \, \Phi(s), \quad \forall s \in \mathcal{S}.
\tag{15}
$$

The proof of Theorem 3.5 can be found in Appendix A.4. Under the update-wise frozen-anchor interpretation above, the theorem shows that the potential-based shaping term does not alter the optimal action, since $\arg\max_a Q_{\mathcal{M}'}^*(s, a) = \arg\max_a Q_{\mathcal{M}}^*(s, a)$. This result supports the use of TEB's intrinsic reward to incorporate predictive bisimulation metric-based novelty into value learning without changing the underlying optimal policy.

## 4. Experiment

In this section, we first compare TEB against powerful visual RL baselines on the challenging MetaWorld benchmark. We then evaluate its pure exploration ability on the Maze2D environment by comparing with recent exploration algorithms. Finally, we present qualitative visualizations and extensive ablation studies to further analyze the key components of TEB.

### 4.1. Experimental Setup

**Environments.** MetaWorld is a challenging benchmark suite for visual robotic manipulation, detailed in the Appendix C.1. In the setting, the agent acts from high-dimensional observations with distracting visual elements, and receives sparse and even success-based rewards. Maze2D is a low-dimensional continuous-control navigation benchmark with 2D maze layouts ranging from simple Corridors to hard Bottleneck task. The source code will be released later.

**Baselines.** In MetaWorld experiments, we conduct a broad comparison of TEB with the current strong visual RL algorithms: DrM (Xu et al., 2024), CTRL-SR (Gao et al., 2025), EME (Wang et al., 2024a), LIBERTY (Wang et al., 2023) and RAP (Chen & Pan, 2022), and the representative exploration algorithms: ICM (Pathak et al., 2017), CeSD (Bai et al., 2024), and LBS (Mazzaglia et al., 2022). All algorithms are built on top of the DrQ-v2 framework

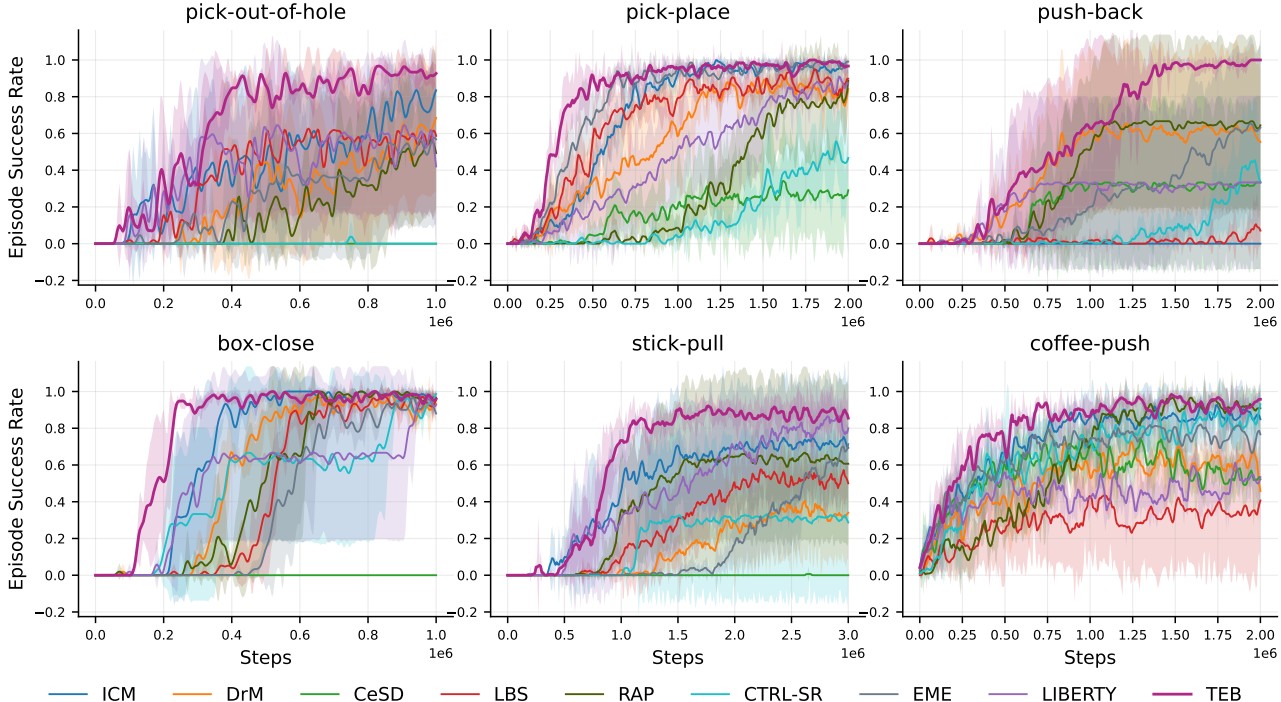

*Figure 2.* Success rates of TEB and baselines in the MetaWorld environment. Each experiment runs with three random seeds, with shaded region representing the standard deviation across seeds.

(Yarats et al., 2021a) except for CTRL-SR that follows the official framework. Specifically, CTRL-SR is the latest algorithm that alleviates representation and exploration difficulties through spectral representations. LIBERTY and EME are two metric-based exploration methods, which respectively introduce heuristic inverse dynamic and policy KL-divergence terms into bisimulation metrics. DrM facilitates policy learning in sparse reward environments by minimizing dormant ratios with visual settings. RAP is a strong algorithm with bisimulation metric-based representation learning. In the Maze environment, the comparative intrinsic exploration baselines consist of common exploration algorithms: LBS (Mazzaglia et al., 2022), RND (Burda et al., 2018), Disagreement (Pathak et al., 2019), ICM (Pathak et al., 2017), and ProtoRL (Yarats et al., 2021b), and 5 skill discovery baselines: CeSD (Bai et al., 2024), BeCL (Yang et al., 2023), LSD (Park et al., 2022), DIAYN (Eysenbach et al., 2018), and SMM (Lee et al., 2019), where CeSD is a recent strong baseline. More details on these baselines in the Appendix C.6.

**Comparison Metrics.** In MetaWorld, the success rate of each task is the metric for comparing all algorithms. We train for different environment steps according to the difficulty of the task. In Maze2D, to quantify exploration ability, we compare the state coverage ratios of the policies over 100K environment steps, where the ratio measures the proportion of visited units to all units in a task map.

### 4.2. MetaWorld Experiments

The experiments evaluate the ability of TEB in enabling task-aware exploration and learning effective policies from pixel observations. We consider six tasks of varying difficulty, e.g., hard *Stick-pull* and middle *Box-close*. Note that even for the simpler *Box-close*, learning an efficient policy remains non-trivial due to its sparse reward structure and significant visual distractions. As shown in Figure 2, TEB consistently outperforms strong baseline methods across all tasks, which shows both faster convergence and higher success rates. In particular, on challenging tasks such as *Stick-pull* and *Push-back*, TEB achieves success rates of 87.9% and 98.4% (the best success rates are in Table 2 of the Appendix C.2) and significantly surpasses all baselines, which empirically demonstrates its strong comprehensive performance in complex visual environments with sparse rewards. By contrast, among intrinsic exploration baselines, we observe that CeSD fails to obtain rewards and successfully complete tasks such as *Push-back*. This is likely due to the limitations of skill discovery–driven intrinsic rewards in challenging visual tasks. It is worth noting that the inverse dynamics model of ICM can also effectively filter out interfering elements, potentially providing strong support for more accurate exploration. In addition, as important comparative baselines, LIBERTY and EME performed significantly worse than the TEB method, and their performance varied greatly across different tasks, such as *Pick-place* and *Stick-*

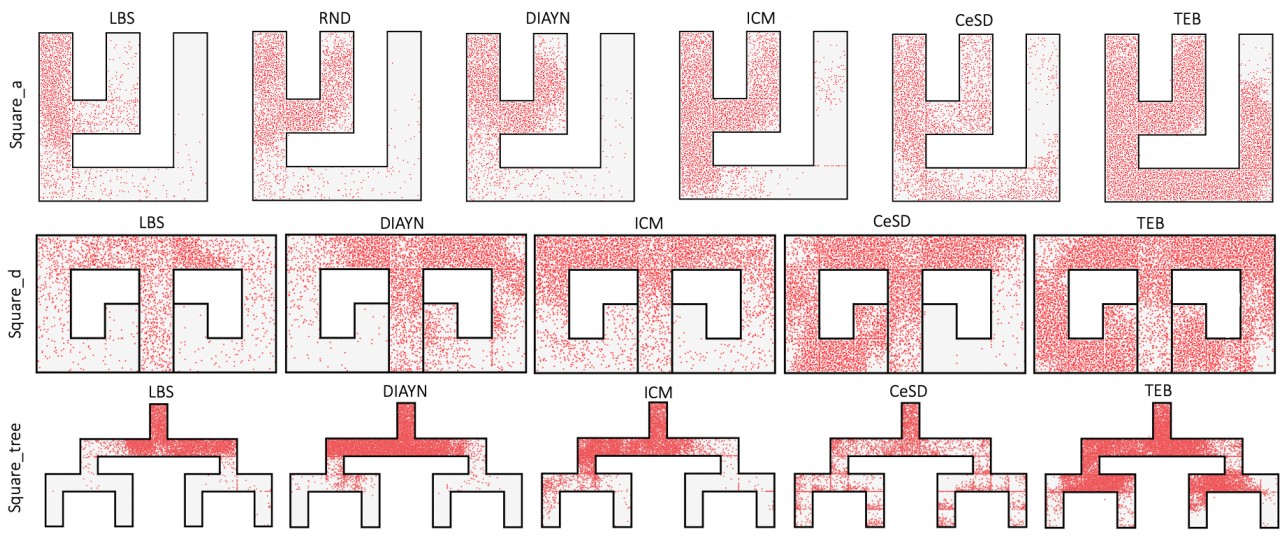

*Figure 3.* Visualization of state coverage in representative Maze2D tasks.

*Table 1.* State coverage ratios in the Maze environment. We report the mean and std of 10 seeds for each algorithm.

| Domains | Square-a | Square-b | Square-c | Square-d | Corridor2 | Square-tree | Square-bottleneck |
|---|---|---|---|---|---|---|---|
| Disagreement | $0.38 \pm 0.08$ | $0.38 \pm 0.20$ | $0.39 \pm 0.19$ | $0.43 \pm 0.14$ | $0.48 \pm 0.10$ | $0.32 \pm 0.10$ | $0.34 \pm 0.07$ |
| ICM | $0.54 \pm 0.08$ | $0.57 \pm 0.14$ | $0.46 \pm 0.06$ | $0.59 \pm 0.05$ | $0.75 \pm 0.07$ | $0.41 \pm 0.04$ | $0.33 \pm 0.06$ |
| LBS | $0.30 \pm 0.04$ | $0.27 \pm 0.05$ | $0.25 \pm 0.02$ | $0.33 \pm 0.03$ | $0.44 \pm 0.07$ | $0.23 \pm 0.04$ | $0.21 \pm 0.04$ |
| Proto | $0.40 \pm 0.04$ | $0.40 \pm 0.06$ | $0.38 \pm 0.09$ | $0.48 \pm 0.04$ | $0.71 \pm 0.04$ | $0.24 \pm 0.02$ | $0.23 \pm 0.01$ |
| RND | $0.42 \pm 0.10$ | $0.60 \pm 0.13$ | $0.39 \pm 0.12$ | $0.37 \pm 0.04$ | $0.63 \pm 0.10$ | $0.28 \pm 0.09$ | $0.32 \pm 0.09$ |
| LIBERTY | $0.62 \pm 0.16$ | $0.46 \pm 0.09$ | $0.47 \pm 0.09$ | $0.57 \pm 0.13$ | $0.77 \pm 0.11$ | $0.41 \pm 0.05$ | $0.35 \pm 0.04$ |
| EME | $0.61 \pm 0.06$ | $0.53 \pm 0.07$ | $0.51 \pm 0.09$ | $0.59 \pm 0.13$ | $0.90 \pm 0.08$ | $0.41 \pm 0.13$ | $0.42 \pm 0.07$ |
| BeCL | $0.52 \pm 0.05$ | $0.48 \pm 0.12$ | $0.43 \pm 0.09$ | $0.47 \pm 0.05$ | $0.67 \pm 0.13$ | $0.37 \pm 0.07$ | $0.30 \pm 0.05$ |
| CeSD | $0.71 \pm 0.05$ | $0.66 \pm 0.05$ | $0.60 \pm 0.05$ | $0.57 \pm 0.06$ | $0.82 \pm 0.06$ | $0.40 \pm 0.02$ | $0.46 \pm 0.05$ |
| LSD | $0.42 \pm 0.03$ | $0.43 \pm 0.06$ | $0.37 \pm 0.02$ | $0.45 \pm 0.03$ | $0.56 \pm 0.05$ | $0.28 \pm 0.02$ | $0.35 \pm 0.04$ |
| DIAYN | $0.43 \pm 0.05$ | $0.48 \pm 0.06$ | $0.42 \pm 0.04$ | $0.47 \pm 0.03$ | $0.57 \pm 0.06$ | $0.37 \pm 0.04$ | $0.28 \pm 0.04$ |
| SMM | $0.42 \pm 0.10$ | $0.35 \pm 0.14$ | $0.32 \pm 0.07$ | $0.35 \pm 0.02$ | $0.84 \pm 0.04$ | $0.25 \pm 0.02$ | $0.34 \pm 0.06$ |
| TEB (Ours) | $\mathbf{0.87 \pm 0.07}$ | $\mathbf{0.85 \pm 0.07}$ | $\mathbf{0.74 \pm 0.04}$ | $\mathbf{0.77 \pm 0.04}$ | $\mathbf{0.93 \pm 0.02}$ | $\mathbf{0.50 \pm 0.04}$ | $\mathbf{0.47 \pm 0.03}$ |

*pull* of EME. Furthermore, CTRL-SR converges slowly in many tasks, such as the *Push-back* task, and may even fail to learn a responsive policy. Note that we followed existing work and performed at least three replicate experiments on all tasks. Despite this number limitation, the experimental results are still able to empirically distinguish the overall performance of different methods across various tasks. Furthermore, we will conduct more extensive and in-depth evaluations in the Maze2D environment described below.

### 4.3. Reward-free Maze2D Experiments

In reward-free Maze2D tasks, our experiments aim to verify whether intrinsic rewards built on the predictive bisimulation metric can demonstrate highly efficient exploration abilities under reward-free settings. Note that since the Maze2D environment directly provides low-dimensional states, TEB does not train representation learning loss $\mathcal{L}_{\text{bisim}}$, but instead

directly measures a metric-based exploration bonus with states. Additionally, in the reward-free tasks, the Gaussian reward predictor still uses the sparse external reward as its reward target.

As shown in Table 1, compared to existing exploration algorithms, TEB exhibits the strongest pure exploration capability, achieving the highest state coverage across various maze layouts, from simple corridors to complex bottlenecks. Specifically, using 100K steps as a benchmark, TEB shows a significant advantage over the previous best baseline algorithm, i.e., CeSD, in square maze series. For example, TEB achieves coverage scores of 0.87 and 0.85 on *Square_a* and *Square_b*, outperforming 0.71 and 0.66 of CeSD, respectively. For similar metric-based baselines, EME and LIBERTY are still inferior to TEB overall. Furthermore, the coverage ratios of other exploration and skill discovery algorithms are significantly lower than those of our TEB.

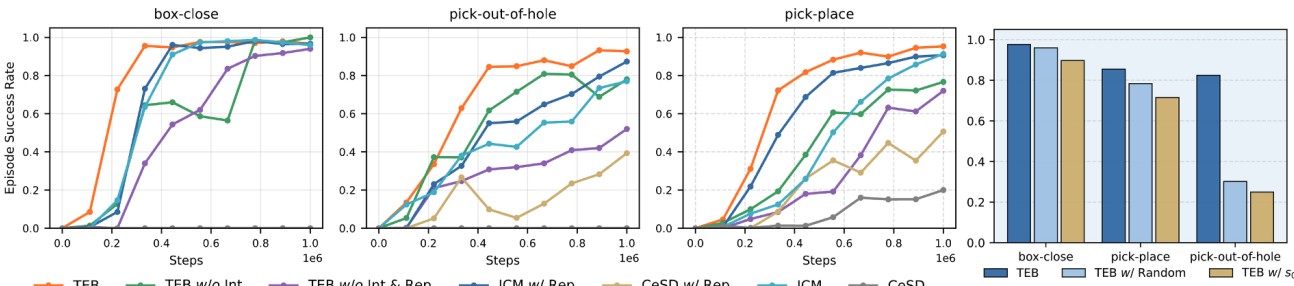

*Figure 4.* Ablation Studies. **Left**: the three figures show the ablation curves of TEB components across three tasks; **Right**: the fourth figure illustrates the ablation results regarding anchor state selection for intrinsic rewards across the three tasks. Each task is run for 1M steps with three random seeds. Note that for better performance comparison, the anchor ablation experiment (right) exhibits the episode success rate at 50% of the training steps.

In addition, detailed comparisons of training curves can be found in Figure 7 of the Appendix C.4. In summary, the experimental results on pure exploration tasks strongly demonstrate that global intrinsic rewards derived from our robust bisimulation metric can efficiently enhance a policy's ability to explore unknown regions. This ability will accelerate the discovery of extrinsic rewards and promote faster policy convergence in practical tasks.

As shown in Figure 8, we also visualize state coverage over 100K training steps by rendering whether each discretized spatial bin of the maze map has been visited. We only show representative tasks for a subset of baselines, and more visualizations are available in the Appendix C.5. Overall, the visualization comparisons intuitively show that TEB achieved the highest map coverage, which qualitatively validates its strongest exploration ability. Specifically, most baselines can only explore areas near the initial state, typically stalling at turns, such as LBS and RND in the *Square_a* and *Square_tree* map. Moreover, compared with the strong CeSD baseline, TEB maintains deep exploration while also stably expanding into the surrounding regions around newly discovered states, such as the *Square_tree* map. These properties indicate that metric-based intrinsic rewards do not prematurely drive value convergence and cause policy stagnation, but instead provide a stable incentive that sustains effective exploration throughout training.

### 4.4. Ablation Studies

**TEB Components.** This ablation study analyzes the effectiveness of predictive bisimulation metric-based representation learning (labeled "Rep") and metric-based intrinsic reward ("Int") in TEB. Therefore, we progressively remove components ("TEB w/o Rep", "TEB w/o Rep & Int") and also plug our representation module into ICM and CeSD ("ICM w/ Rep", "CeSD w/ Rep"). As depicted in Figure 4, we observe several intuitive results. (1) Comparing the first three variants in the figure, both the representation module and the intrinsic module provide non-trivial gains on all tasks except *Box-close*, which validates the effectiveness

of each component. (2) Comparing TEB against powerful "ICM w/ Rep" and "CeSD w/ Rep", we find that even under a unified metric-based representation backbone, TEB still achieves the best overall performance, which also shows the effectiveness of our intrinsic module and implies the systematic gains from the task-aware exploration. (3) Comparing the last four, plugging our representation into ICM and CeSD consistently improves their performance, suggesting that the module is broadly beneficial. These results provide strong evidence that both the representation and intrinsic reward components of TEB are individually effective and more importantly, yield powerful performance when coupled within a metric-based framework.

**Anchor State Selection.** As shown in Figure 4, the right bar chart reports the episode success rate of the three selection strategies of anchor states. Our TEB with the pseudo-anchor state outperforms both the ("TEB w/ Random") with random anchor states and the ("TEB w/ $s_0$") with fixed initial-state anchors, i.e., $s_0$, across all tasks, especially in *Pick-out-of-hole*. Furthermore, we directly observe that suboptimal random anchor strategy (random permutations from batches) is slightly better than the fixed initial anchor strategy, but not significantly. Overall, these results highlight that the anchor selection is critical, and demonstrate the superiority of our anchor strategy.

### 4.5. Effectiveness of Predicted Gaussian Rewards

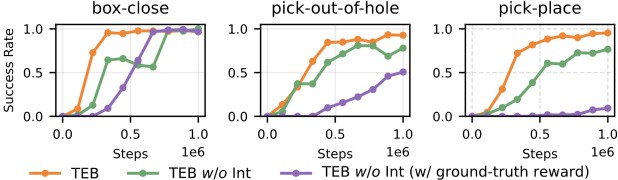

*Figure 5.* Performance analysis of the bisimulation metric with different reward signals.

As shown in Figure 5, we consider two metric-based representations without intrinsic exploration based on TEB, where the internal reward differentials (in Eq. (5)) use pre-

dicted Gaussian rewards and ground-truth rewards from the environment, respectively. Comparative results across three tasks show that our metric-based representations with predicted rewards significantly outperform the traditional environment rewards on most tasks. Furthermore, the "TEB w/o Int" curve reveals that the predicted method can learn quickly. In summary, combined with the TEB curve, the final results indicate that the performance improvement of TEB stems not only from the predicted Gaussian reward but also from the downstream metric-based intrinsic reward, which provides a considerable performance boost.

## 5. Related Work

Reinforcement learning has made notable progress in intrinsic exploration (Pathak et al., 2019; Park et al., 2022; Yang et al., 2023; Mazzaglia et al., 2022; Zheng et al., 2024; Sun et al., 2025). Early approaches like the intrinsic curiosity module (Pathak et al., 2017) promote exploration by leveraging dynamic prediction error. The CIC (Laskin et al., 2022) introduces a contrastive objective to maximize the mutual information between latent skills and resulting trajectories. The recent work(Bai et al., 2024) diversifies skills using policy ensembles regularized by entropy constraints, while preserving behavioral consistency. Variants like DVFB (Sun et al., 2025) enable skill discovery with strong generalization by aligning policies across time. METRA (Park et al., 2024) learns metric-aware abstractions for unsupervised RL by maximizing temporal distances, but it remains fundamentally task-agnostic. ETD (Jiang et al., 2025) measures episodic novelty through temporal distance and therefore favors temporally distant states, yet temporal distance alone may still overvalue regions that are far in time but irrelevant to downstream task progress. Furthermore, most of these methods are tested in low-dimensional (Jiang et al., 2022; Wang et al., 2024b), highly task-specific tasks and are often difficult to extend to visual tasks (Laskin et al., 2022).

Visual RL under sparse rewards remains challenging due to both the variations and the weak reward signals (Yarats et al., 2021b; Zheng et al., 2023; Mete et al., 2024). Most recent work focuses on visual representation learning (Li et al., 2024; Lee & Hwang, 2025; Zhang et al., 2025), with limited efforts by coupled it with reward explorations (Yarats et al., 2021b). Specifically, the methods widely adopt bisimulation metrics (Castro et al., 2021), mutual information maximization (Cao et al., 2025), and contrastive augmentation (Laskin et al., 2020) to learn task-relevant latent spaces. DrM (Xu et al., 2024) boosts representation and exploration via neuron activation, but its task-agnostic bonus demands careful hyperparameter tuning. Additionally, spectral representation (Gao et al., 2025) leverages spectral decomposition of the transition operator to capture compact dynamic features. More closely related to our work are metric-based explo-

ration methods. LIBERTY (Wang et al., 2023) and EME (Wang et al., 2024a) construct exploration bonuses from bisimulation-style metrics, but they respectively augment the metric with heuristic inverse dynamic and policy KL-divergence terms. In contrast, TEB is built on the standard bisimulation form, i.e., a reward differential together with a transition differential, thus preserving the bisimulation metric properties and theory. Importantly, TEB proposes the predictive reward differential and the global anchor-state mechanism to explicitly address the problems of metric degeneration and local exploration.

## 6. Conclusion

We have introduced a task-aware exploration framework (TEB) that tightly couples visual representation and exploration through a robust predictive bisimulation metric. TEB keeps the metric non-degenerate using predicted Gaussian rewards under sparse rewards, and then builds an effective metric-based intrinsic reward to drive task-aware global exploration without changing the optimal policy. Experiments on challenging MetaWorld and Maze2D show TEB significantly boosts the exploration ability and policy performance.

## Acknowledgements

This work was supported by the National Natural Science Foundation of China (Grant No. 61772438 and No. 61375077) and the China Scholarship Council.

## Impact Statement

This paper presents work whose goal is to advance the field of machine learning. There are many potential societal consequences of our work, none of which we feel must be specifically highlighted here.

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

# A. Theoretical Analysis

## A.1. The Proof of the Convergence and Fixed Point

**Lemma A.1** (Convergence and Fixed Point). *Predictive bisimulation metric $\widetilde{d}_{pre}$ is a contraction mapping w.r.t the $L^\infty$ norm on $\mathbb{R}^{\mathcal{S} \times \mathcal{S}}$, and there exists a fixed point $\widetilde{d}_{pre}$.*

*Proof.* We refer to the work of Kemertas et al. (Kemertas & Aumentado-Armstrong, 2021) to prove this. Take any $d, d' \in \mathcal{M}$ and any state pair $(s_i, s_j)$. By the operator definition in Eq. (5), the predictive reward differential $\Delta_R^\pi(s_i, s_j)$ does not depend on $d$, hence it cancels:

$$
\begin{aligned}
&\left| (\widehat{\mathcal{T}}^\pi d)(s_i, s_j) - (\widehat{\mathcal{T}}^\pi d')(s_i, s_j) \right| \\
&= \left| \Big( c_R \Delta_R^\pi(s_i, s_j) + c_T \mathcal{W}_1(d)\big(\widehat{\mathcal{P}}^\pi(\cdot \mid s_i), \widehat{\mathcal{P}}^\pi(\cdot \mid s_j)\big) \Big) - \Big( c_R \Delta_R^\pi(s_i, s_j) + c_T \mathcal{W}_1(d')\big(\widehat{\mathcal{P}}^\pi(\cdot \mid s_i), \widehat{\mathcal{P}}^\pi(\cdot \mid s_j)\big) \Big) \right| \\
&= \left| \Big( c_R \Delta_R^\pi(s_i, s_j) - c_R \Delta_R^\pi(s_i, s_j) \Big) + \Big( c_T \mathcal{W}_1(d)\big(\widehat{\mathcal{P}}^\pi(\cdot \mid s_i), \widehat{\mathcal{P}}^\pi(\cdot \mid s_j)\big) - c_T \mathcal{W}_1(d')\big(\widehat{\mathcal{P}}^\pi(\cdot \mid s_i), \widehat{\mathcal{P}}^\pi(\cdot \mid s_j)\big) \Big) \right| \\
&= c_T \left| \mathcal{W}_1(d)\big(\widehat{\mathcal{P}}^\pi(\cdot \mid s_i), \widehat{\mathcal{P}}^\pi(\cdot \mid s_j)\big) - \mathcal{W}_1(d')\big(\widehat{\mathcal{P}}^\pi(\cdot \mid s_i), \widehat{\mathcal{P}}^\pi(\cdot \mid s_j)\big) \right|.
\end{aligned}
\tag{16}
$$

Let $\mu := \widehat{\mathcal{P}}^\pi(\cdot | s_i)$ and $\nu := \widehat{\mathcal{P}}^\pi(\cdot | s_j)$. Using the primal form of the 1-Wasserstein distance induced by a ground metric $d$,

$$
\mathcal{W}_1(d)(\mu, \nu) = \inf_{\zeta \in \Pi(\mu, \nu)} \mathbb{E}_{(s', \bar{s}') \sim \zeta}\big[ d(s', \bar{s}') \big] = \inf_{\zeta \in \Pi(\mu, \nu)} \sum_{s', \bar{s}'} \zeta(s', \bar{s}') \, d(s', \bar{s}'),
\tag{17}
$$

where $\Pi(\mu, \nu)$ is the set of couplings with marginals $\mu$ and $\nu$. Then we can bound the difference between the two Wasserstein terms as

$$
\begin{aligned}
\left| \mathcal{W}_1(d)(\mu, \nu) - \mathcal{W}_1(d')(\mu, \nu) \right| &= \left| \inf_{\zeta \in \Pi(\mu, \nu)} \mathbb{E}_\zeta[d] - \inf_{\zeta \in \Pi(\mu, \nu)} \mathbb{E}_\zeta[d'] \right| \\
&\leq \sup_{\zeta \in \Pi(\mu, \nu)} \left| \mathbb{E}_\zeta[d - d'] \right| \\
&\leq \sup_{\zeta \in \Pi(\mu, \nu)} \mathbb{E}_\zeta\big[ |d - d'| \big] \ \leq \ \|d - d'\|_\infty.
\end{aligned}
\tag{18}
$$

Combining Eq. (16) and Eq. (18) yields

$$
\left| (\widehat{\mathcal{T}}^\pi d)(s_i, s_j) - (\widehat{\mathcal{T}}^\pi d')(s_i, s_j) \right| \leq c_T \|d - d'\|_\infty.
\tag{19}
$$

Taking the supremum over $(s_i, s_j) \in \mathcal{S} \times \mathcal{S}$ gives

$$
\|\widehat{\mathcal{T}}^\pi d - \widehat{\mathcal{T}}^\pi d'\|_\infty \leq c_T \|d - d'\|_\infty.
\tag{20}
$$

Since $c_T \in [0, 1)$, $\widehat{\mathcal{T}}^\pi$ is a contraction on $(\mathcal{M}, \|\cdot\|_\infty)$. By the Banach fixed-point theorem, there exists a unique fixed point $\widetilde{d}_{\text{pre}} \in \mathcal{M}$ such that $\widetilde{d}_{\text{pre}} = \widehat{\mathcal{T}}^\pi \widetilde{d}_{\text{pre}}$. $\square$

## A.2. The Proof of the Predictive Reward Prevents Degenerate Metric

**Theorem A.2** (Predictive Reward Prevents Degenerate Metric). *Let the state space $\mathcal{S}$ be compact and the policy $\pi$ fixed. Consider the predictive bisimulation operator $\widehat{\mathcal{T}}^\pi$, the learned reward model satisfies $\hat{r}_\theta(s_t, a_t) = r_t + \varepsilon_t$ with $\mathrm{Var}[\varepsilon_t | s_t, a_t] = \Sigma > 0$. Then, the unique fixed point $\widetilde{d}_{pre} = \widehat{\mathcal{T}}^\pi \widetilde{d}_{pre}$ admits a strictly positive expected diameter in the sparse-reward region $\mathcal{S}_0 = \{s | r(s, a) = 0\}$:*

$$
\mathbb{E}_{\mathcal{D}}\big[ \mathrm{diam}(\mathcal{S}_0; \widetilde{d}_{pre}) \big] \in \left( 0, \frac{c_R}{1 - c_T} \cdot \frac{2}{\sqrt{\pi}} \sqrt{\Sigma} \right].
\tag{21}
$$

.

*Proof.* Fix $\pi$ and consider $(s_i, s_j) \in \mathcal{S}_0 \times \mathcal{S}_0$ sampled from $\mathcal{D}$. By the fixed-point equation of $\tilde{d}_{\text{pre}}$,

$$\tilde{d}_{\text{pre}}(s_i, s_j) = c_R \Delta_R^\pi(s_i, s_j) + c_T \mathcal{W}_1(\tilde{d}_{\text{pre}})(\mathcal{P}^\pi(\cdot|s_i), \mathcal{P}^\pi(\cdot|s_j)), \tag{22}$$

where $\Delta_R^\pi(s_i, s_j) = \mathbb{E}_{a_i, a_j \sim \pi}\big[\big|\hat{r}_\theta(s_i, a_i) - \hat{r}_\theta(s_j, a_j)\big|\big]$. Since $\mathcal{W}_1(\cdot) \geq 0$ and $\Delta_R^\pi \geq 0$, we immediately have

$$\mathbb{E}_{\mathcal{D}}\big[\tilde{d}_{\text{pre}}(s_i, s_j)\big] \geq c_R \mathbb{E}_{\mathcal{D}}\big[\Delta_R^\pi(s_i, s_j)\big]. \tag{23}$$

For the upper bound, take $\mathbb{E}_{\mathcal{D}}[\cdot]$ on both sides of Eq. (22). Using the primal form of $W_1$ and choosing the product coupling gives $W_1(\tilde{d}_{\text{pre}})(\mu, \nu) \leq \mathbb{E}_{x \sim \mu, y \sim \nu}[\tilde{d}_{\text{pre}}(x, y)]$. The expected cost under this product coupling is bounded by the same pairwise expectation, hence

$$\mathbb{E}_{\mathcal{D}}\Big[\mathcal{W}_1(\tilde{d}_{\text{pre}})(\mathcal{P}^\pi(\cdot|s_i), \mathcal{P}^\pi(\cdot|s_j))\Big] \leq \mathbb{E}_{\mathcal{D}}\big[\tilde{d}_{\text{pre}}(s_i, s_j)\big]. \tag{24}$$

Substituting Eq. (24) into the expectation of Eq. (22) yields

$$\begin{aligned}\mathbb{E}_{\mathcal{D}}\big[\tilde{d}_{\text{pre}}(s_i, s_j)\big] &\leq c_R \mathbb{E}_{\mathcal{D}}\big[\Delta_R^\pi(s_i, s_j)\big] + c_T \mathbb{E}_{\mathcal{D}}\big[\tilde{d}_{\text{pre}}(s_i, s_j)\big], \\ &\leq \frac{c_R}{1 - c_T} \mathbb{E}_{\mathcal{D}}\big[\Delta_R^\pi(s_i, s_j)\big],\end{aligned} \tag{25}$$

It remains to bound $\mathbb{E}_{\mathcal{D}}[\Delta_R^\pi]$ on $\mathcal{S}_0$. For $s \in \mathcal{S}_0$ we have $r(s, a) = 0$, so the learned reward model satisfies $\hat{r}_\theta(s, a) = r(s, a) + \varepsilon = \varepsilon$ with $\text{Var}[\varepsilon \mid s, a] = \Sigma > 0$. Therefore, for $(s_i, s_j) \in \mathcal{S}_0 \times \mathcal{S}_0$, $\hat{r}_\theta(s_i, a_i) - \hat{r}_\theta(s_j, a_j) = \varepsilon_i - \varepsilon_j \triangleq \delta$, where $\delta$ is a non-degenerate Gaussian. If $\varepsilon_i$ and $\varepsilon_j$ are conditionally independent given $(s_i, a_i)$ and $(s_j, a_j)$, then $\text{Var}[\delta] = 2\Sigma$, and hence $\delta \sim \mathcal{N}(0, 2\Sigma)$. Let $z \sim \mathcal{N}(0, 1)$, then $\delta = \sqrt{2\Sigma} z$ and

$$\mathbb{E}\big[|\delta|\big] = \sqrt{2\Sigma} \, \mathbb{E}\big[|z|\big] = \sqrt{2\Sigma} \cdot \sqrt{\frac{2}{\pi}} = \frac{2}{\sqrt{\pi}} \sqrt{\Sigma}. \tag{26}$$

Moreover, $\mathbb{E}[|\delta|] > 0$ since $\Sigma > 0$. Therefore, on $\mathcal{S}_0$ we have

$$\begin{aligned}\mathbb{E}_{\mathcal{D}}[\Delta_R^\pi(s_i, s_j)] &= \mathbb{E}_{\mathcal{D}} \mathbb{E}_{a_i, a_j \sim \pi}\Big[\big|\hat{r}_\theta(s_i, a_i) - \hat{r}_\theta(s_j, a_j)\big|\Big] \\ &= \mathbb{E}\big[|\delta|\big] \in \left(0, \frac{2}{\sqrt{\pi}}\sqrt{\Sigma}\right].\end{aligned} \tag{27}$$

Combining this with Eq. (23), Eq. (25) and Eq. (27) gives

$$0 < \mathbb{E}_{\mathcal{D}}\Big[\tilde{d}_{\text{pre}}(s_i, s_j)\Big] \leq \frac{c_R}{1 - c_T} \cdot \frac{2}{\sqrt{\pi}}\sqrt{\Sigma}. \tag{28}$$

By definition of $\text{diam}(\mathcal{S}_0; \tilde{d}_{\text{pre}})$ as the expected pairwise distance over $\mathcal{S}_0 \times \mathcal{S}_0$ under $\mathcal{D}$, we proved it. $\qquad\square$

### A.3. The Proof of the Value Difference Bound

**Theorem A.3** (Value Difference Bound). *Given states $s_i$ and $s_j$, and a policy $\pi$, let $V^\pi(s)$ be the state value function over policy $\pi$, we have,*

$$\big|V^\pi(s_i) - V^\pi(s_j)\big| \leq \tilde{d}_{\text{pre}}(s_i, s_j). \tag{11}$$

*Proof.* We prove the theorem by induction based on the work of Chen et al. (Chen & Pan, 2022). We assume the predictive reward random variable $\hat{r}_\theta(s, a)$ satisfies $\mathbb{E}[\hat{r}_\theta(s, a)] = r(s, a)$ (equivalently, $\hat{r}_\theta(s, a) = r(s, a) + \varepsilon$ with $\mathbb{E}[\varepsilon|s, a] = 0$), and we use the transition $\mathcal{P}^\pi(\cdot|s) = \mathbb{E}_{a \sim \pi(\cdot|s)} \mathcal{P}(\cdot|s, a)$.

Firstly, we provide the following Bellman and metric iterate forms, as well as a key inequality. We define the standard Bellman evaluation operator with the environment reward $r(s, a)$,

$$(\mathcal{T}^\pi V)(s) := \mathbb{E}_{a \sim \pi(\cdot|s)}\big[r(s, a)\big] + \gamma \, \mathbb{E}_{s' \sim \mathcal{P}^\pi(\cdot|s)}\big[V(s')\big], \tag{29}$$

with $V_0^\pi(\cdot) \equiv 0$ and $V_{n+1}^\pi = \mathcal{T}^\pi V_n^\pi$.

We consider the predictive bisimulation operator defined in Eq. (5):

$$(\mathcal{T}_b^\pi d)(s_i, s_j) := c_R \Delta_R^\pi(s_i, s_j) + c_T \, \mathcal{W}_1(d)(\mathcal{P}^\pi(\cdot|s_i), \mathcal{P}^\pi(\cdot|s_j)), \tag{30}$$

where

$$\Delta_R^\pi(s_i, s_j) := \mathbb{E}_{a_i \sim \pi(\cdot|s_i), \, a_j \sim \pi(\cdot|s_j)}\Big[\big|\hat{r}_\theta(s_i, a_i) - \hat{r}_\theta(s_j, a_j)\big|\Big], \tag{31}$$

with $d_0 \equiv 0$ and $d_{n+1} = \mathcal{T}_b^\pi d_n$. By contraction, $V_n^\pi \to V^\pi$ and $d_n \to \tilde{d}_{\text{pre}}$.

Then, we derive a key inequality for the transition term. Assume a function $f$ satisfies $|f(x) - f(y)| \le d(x, y)$ for all $x, y$. For any coupling $\zeta \in \Pi(\mu, \nu)$,

$$\left|\mathbb{E}_{x \sim \mu}[f(x)] - \mathbb{E}_{y \sim \nu}[f(y)]\right| = \left|\mathbb{E}_{(x,y) \sim \zeta}\big[f(x) - f(y)\big]\right| \tag{32}$$

$$\le \mathbb{E}_{(x,y) \sim \zeta}\big|f(x) - f(y)\big|$$

$$\le \mathbb{E}_{(x,y) \sim \zeta} d(x, y).$$

Taking $\inf_{\zeta \in \Pi(\mu, \nu)}$ yields,

$$\left|\mathbb{E}_{x \sim \mu}[f(x)] - \mathbb{E}_{y \sim \nu}[f(y)]\right| \le W_1(d)(\mu, \nu). \tag{33}$$

Secondly, we prove by induction that for all $n \ge 0$ and all $s_i, s_j$,

$$\big|V_n^\pi(s_i) - V_n^\pi(s_j)\big| \le d_n(s_i, s_j). \tag{34}$$

The case $n = 0$ holds since both sides are 0.

Assume Eq. (34) holds for some $n$. Then we unfold:

$$\big|V_{n+1}^\pi(s_i) - V_{n+1}^\pi(s_j)\big| = \left|(\mathcal{T}^\pi V_n^\pi)(s_i) - (\mathcal{T}^\pi V_n^\pi)(s_j)\right| \tag{35}$$

$$= \left|\mathbb{E}_{a_i}[r(s_i, a_i)] - \mathbb{E}_{a_j}[r(s_j, a_j)] + \gamma\Big(\mathbb{E}_{s_i'}[V_n^\pi(s_i')] - \mathbb{E}_{s_j'}[V_n^\pi(s_j')]\Big)\right|$$

$$\le \left|\mathbb{E}_{a_i}[r(s_i, a_i)] - \mathbb{E}_{a_j}[r(s_j, a_j)]\right| + \gamma\left|\mathbb{E}_{s_i'}[V_n^\pi(s_i')] - \mathbb{E}_{s_j'}[V_n^\pi(s_j')]\right|.$$

For the first term, using $\mathbb{E}[\hat{r}_\theta(s, a)] = r(s, a)$ and the property of $|\mathbb{E}[X]| \le \mathbb{E}[|X|]$, we have,

$$\left|\mathbb{E}_{a_i}[r(s_i, a_i)] - \mathbb{E}_{a_j}[r(s_j, a_j)]\right| = \left|\mathbb{E}_{a_i}\mathbb{E}[\hat{r}_\theta(s_i, a_i)] - \mathbb{E}_{a_j}\mathbb{E}[\hat{r}_\theta(s_j, a_j)]\right| \tag{36}$$

$$= \left|\mathbb{E}_{a_i, a_j}\mathbb{E}\big[\hat{r}_\theta(s_i, a_i) - \hat{r}_\theta(s_j, a_j)\big]\right|$$

$$\le \mathbb{E}_{a_i, a_j}\mathbb{E}\Big[\big|\hat{r}_\theta(s_i, a_i) - \hat{r}_\theta(s_j, a_j)\big|\Big]$$

$$= \Delta_R^\pi(s_i, s_j). \tag{37}$$

For the second term, the induction hypothesis implies $|V_n^\pi(x) - V_n^\pi(y)| \le d_n(x, y)$ for all $x, y$, hence applying Eq. (33) with $f = V_n^\pi, d = d_n, \mu = \mathcal{P}^\pi(\cdot|s_i)$ and $\nu = \mathcal{P}^\pi(\cdot|s_j)$ gives that,

$$\left|\mathbb{E}_{s_i'}[V_n^\pi(s_i')] - \mathbb{E}_{s_j'}[V_n^\pi(s_j')]\right| \le \mathcal{W}_1(d_n)(\mathcal{P}^\pi(\cdot|s_i), \mathcal{P}^\pi(\cdot|s_j)). \tag{38}$$

Substituting Eq. (37) and Eq. (38) into Eq. (36), we have,

$$\big|V_{n+1}^\pi(s_i) - V_{n+1}^\pi(s_j)\big| \le c_R \Delta_R^\pi(s_i, s_j) + c_T \, \mathcal{W}_1(d_n)(\mathcal{P}^\pi(\cdot|s_i), \mathcal{P}^\pi(\cdot|s_j)) = d_{n+1}(s_i, s_j), \tag{39}$$

so Eq. (34) holds for $n + 1$.

Finally, letting $n \to \infty$ in Eq. (34) and using $V_n^\pi \to V^\pi$ and $d_n \to \tilde{d}_{\text{pre}}$ yields

$$\big|V^\pi(s_i) - V^\pi(s_j)\big| \le \tilde{d}_{\text{pre}}(s_i, s_j), \tag{40}$$

which proves Eq. (11). $\qquad\square$

### A.4. The Proof of the Policy Invariance of the Predictive Bisimulation Shaping

**Theorem A.4** (Policy Invariance of Metric-based Shaping). *Let $\mathcal{M} = (\mathcal{S}, \mathcal{A}, \mathcal{P}, r, \gamma)$ denote the original MDP, and consider the modified reward function $r' = r + \eta\, F(s, a, a')$, where $F$ is defined in Eq. (14). Then the modified MDP $\mathcal{M}' = (\mathcal{S}, \mathcal{A}, \mathcal{P}, r', \gamma)$ with the metric-based potential preserves the optimal policy of $\mathcal{M}$ by:*

$$Q^*_{\mathcal{M}'}(s, a) = Q^*_{\mathcal{M}}(s, a) - \eta\, \Phi(s), \quad \forall s \in \mathcal{S}. \tag{41}$$

*Proof.* Given the potential shaping $F(s_t, a, s') = \gamma\, \Phi(s') - \Phi(s)$ and modified reward $r'(s, a, s') = r(s, a, s') + \eta F(s, a, s')$. According to the Bellman optimality equation, the optimal action-value function of the original MDP $\mathcal{M}$ has the form,

$$Q^*_{\mathcal{M}}(s, a) = \mathbb{E}_{s' \sim \mathcal{P}(\cdot|s,a)}\Big[ r(s, a, s') + \gamma \max_{a'} Q^*_{\mathcal{M}}(s', a') \Big]. \tag{42}$$

We can obtain the following by subtracting the potential function from both sides:

$$
\begin{aligned}
Q^*_{\mathcal{M}}(s, a) - \eta\Phi(s) &= \mathbb{E}_{s' \sim \mathcal{P}(\cdot|s,a)}\Big[ r(s, a, s') + \gamma \max_{a'} Q^*_{\mathcal{M}}(s', a') \Big] - \eta\Phi(s) \\
&= \mathbb{E}_{s' \sim \mathcal{P}(\cdot|s,a)}\Big[ r(s, a, s') + \eta(\gamma\Phi(s') - \Phi(s)) + \gamma \max_{a'} \big( Q^*_{\mathcal{M}}(s', a') - \eta\Phi(s') \big) \Big] \\
&= \mathbb{E}_{s' \sim \mathcal{P}(\cdot|s,a)}\Big[ r(s, a, s') + F(s, a, s') + \gamma \max_{a'} \big( Q^*_{\mathcal{M}}(s', a') - \eta\Phi(s') \big) \Big] \\
&= \mathbb{E}_{s' \sim \mathcal{P}(\cdot|s,a)}\Big[ r'(s, a, s') + \gamma \max_{a'} \big( Q^*_{\mathcal{M}}(s', a') - \eta\Phi(s') \big) \Big].
\end{aligned}
\tag{43}
$$

where we use that $-\eta\Phi(s)$ is independent of $s'$ and

$$\gamma \max_{a'} Q^*_{\mathcal{M}}(s', a') = \gamma \max_{a'} \Big( \big( Q^*_{\mathcal{M}}(s', a') - \eta\Phi(s') \big) + \eta\Phi(s') \Big) = \gamma\eta\Phi(s') + \gamma \max_{a'} \big( Q^*_{\mathcal{M}}(s', a') - \eta\Phi(s') \big). \tag{44}$$

Then, combining Eq. (43) with $\hat{Q}(s, a) \triangleq Q^*_{\mathcal{M}}(s, a) - \eta\Phi(s)$, we have,

$$\hat{Q}(s, a) = \mathbb{E}_{s' \sim \mathcal{P}(\cdot|s,a)}\Big[ r'(s, a, s') + \gamma \max_{a'} \hat{Q}(s', a') \Big], \tag{45}$$

which is exactly the Bellman optimality equation for $Q^*_{\mathcal{M}'}$. By uniqueness of the fixed point, $\hat{Q}(s, a) = Q^*_{\mathcal{M}'}(s, a)$, i.e.,

$$Q^*_{\mathcal{M}'}(s, a) = Q^*_{\mathcal{M}}(s, a) - \eta\Phi(s). \tag{46}$$

Since the shift term $-\eta\Phi(s)$ does not depend on $a$, the maximizing action is unchanged for every $s$, so $\mathcal{M}'$ preserves the optimal policy of $\mathcal{M}$. $\qquad\square$

## B. Extended Preliminaries

### B.1. Intrinsic exploration

Exploration is a key bottleneck in sparse-reward tasks, where extrinsic feedback provides little guidance for discovering reward-relevant trajectories. A common approach augments the environment reward with an intrinsic bonus, i.e., $r'_t = r_t + \eta\, r^{\text{int}}_t$, where $\eta > 0$ controls the contribution of intrinsic motivation. Prior work designs $r^{\text{int}}_t$ from novelty signals, such as information gain, pseudo-count measures, or prediction error in forward models and disagreement among ensembles. While effective in some domains, especially in low-dimensional domains, these can be dominated by task-irrelevant visual factors, leading to inefficient exploration.

### B.2. Backbone framework

In MetaWorld experiments, our method and most of visual RL baselines are implemented on top of the off-policy DrQ-v2 framework, which has become a widely adopted backbone for pixel-based continuous-control benchmarks. Given an encoder $\phi_\omega$, DrQ-v2 trains a double-Q critic $Q_{\vartheta_k}$ using an $n$-step bootstrapped TD loss:

$$\mathcal{L}_Q(\vartheta_k, \phi_\omega) = \mathbb{E}_{\mathcal{D}}\left[ Q_{\vartheta_k}(\phi_\omega(s_t), a_t) - \Big( \sum_{i=0}^{n-1} \gamma^i r_{t+i} + \gamma^n \min_{k=0,1} Q^{\text{tgt}}_{\bar{\vartheta}_k}(\phi_\omega(s_{t+n}), a_{t+n}) \Big) \right]^2, \tag{47}$$

where $Q_{\bar{\vartheta}_k}^{\text{tgt}}$ is a slowly-updated target critic (EMA). The actor $\pi_\nu$ is optimized to maximize the critic value,

$$\mathcal{L}_\pi(\nu) = -\mathbb{E}_\mathcal{D}\left[\min_{k=0,1} Q_{\vartheta_k}\big(\phi_\omega(s_t),\, \pi_\nu(\phi_\omega(s_t)) + \epsilon(t)\big)\right], \tag{48}$$

with exploration noise $\epsilon(t) \sim \text{clip}(\mathcal{N}(0, \sigma_t^2))$. DrQ-v2 further improves sample efficiency by applying lightweight image augmentations, while typically updating $\phi_\omega$ only through the critic.

## C. Experimental Details

### C.1. MetaWorld Tasks

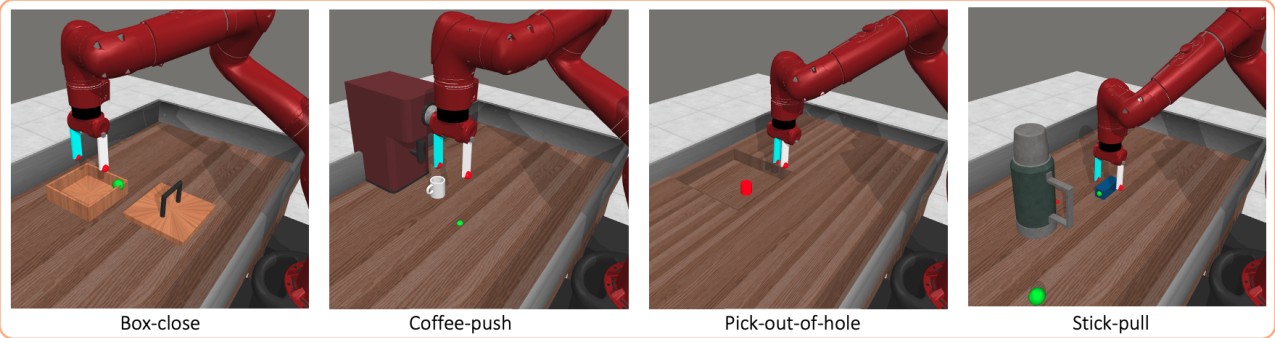

*Figure 6.* Example vision tasks in the MetaWorld environment.

In Figure 6, we show four representative manipulation tasks from the Meta-World benchmark (Yu et al., 2020): *Box-close*, *Stick-pull*, *Pick-out-of-hole*, and *Coffee-push*. These tasks share a common structure in requiring precise, multi-stage object interactions, while differing in contact patterns and control objectives.

- **Box-close** requires the agent to grasp or push a box lid and rotate it until the box is fully closed, demanding accurate spatial alignment and force application.

- **Stick-pull** involves grasping a stick and pulling an object toward a target region, emphasizing directional control through tool-mediated interaction.

- **Pick-out-of-hole** focuses on extracting an object from a cavity, which requires coordinated grasping and vertical lifting under tight spatial constraints.

- **Coffee-push** asks the agent to push a mug to a designated goal position, relying on sustained contact and long-horizon pushing behavior.

Due to the use of high-dimensional pixel-based observations, intrinsic reward mechanisms like prediction error or novelty estimation can become unreliable. Agents may be misled by minor visual fluctuations, such as lighting changes, texture noise, or background shifts, which artificially inflate intrinsic reward signals. As a result, intrinsic motivation methods (e.g., RND or ICM) may overestimate the novelty of visually altered yet semantically identical states. This sensitivity to visual distractions degrades the stability of exploration and hinders efficient policy learning in sparse reward environments.

### C.2. MetaWorld Experiments

Table 2 reports the best episode success rates, in MetaWorld tasks. In general, TEB achieves the highest performance in most tasks, indicating that the proposed metric-based exploration not only accelerates learning, but also achieves a stronger final performance. The gains are most pronounced on harder manipulation tasks where sparse feedback and long-horizon coordination matter: on *Pick-out-of-hole*, TEB reaches $93.1\%$, outperforming the strongest baseline ICM ($76.1\%$) while CeSD and CTRL-SR fail to solve the task; on *Push-back*, TEB attains $98.4\%$, exceeding RAP ($65.0\%$) and DrM ($61.1\%$) by a large margin. TEB also remains consistently strong on *Stick-pull* and *Coffee-push*. The only exception is *Box-close*, where ICM slightly leads ($97.3\%$ vs. $96.8\%$), suggesting that when the task provides relatively easier success feedback, generic curiosity can match peak returns. Notably, the small standard deviations of TEB across all tasks indicate stable convergence.

*Table 2.* We report the best episode success rate of TEB and baselines with mean ± std on MetaWorld tasks.

| Methods | Pick-out-of-hole | Pick-place | Push-back | Box-close | Stick-pull | Coffee-push |
|---|---|---|---|---|---|---|
| ICM | $0.761 \pm 0.045$ | $0.955 \pm 0.014$ | $0.000 \pm 0.000$ | $\mathbf{0.973} \pm 0.013$ | $0.715 \pm 0.016$ | $0.870 \pm 0.019$ |
| DrM | $0.599 \pm 0.065$ | $0.811 \pm 0.030$ | $0.611 \pm 0.024$ | $0.900 \pm 0.028$ | $0.334 \pm 0.025$ | $0.606 \pm 0.046$ |
| CeSD | $0.000 \pm 0.000$ | $0.267 \pm 0.015$ | $0.319 \pm 0.010$ | $0.000 \pm 0.000$ | $0.000 \pm 0.000$ | $0.793 \pm 0.027$ |
| LBS | $0.576 \pm 0.022$ | $0.873 \pm 0.037$ | $0.053 \pm 0.027$ | $0.937 \pm 0.027$ | $0.517 \pm 0.025$ | $0.344 \pm 0.022$ |
| RAP | $0.502 \pm 0.041$ | $0.776 \pm 0.031$ | $0.650 \pm 0.014$ | $0.970 \pm 0.014$ | $0.627 \pm 0.013$ | $0.791 \pm 0.024$ |
| LIBERTY | $0.578 \pm 0.048$ | $0.860 \pm 0.023$ | $0.330 \pm 0.009$ | $0.873 \pm 0.098$ | $0.824 \pm 0.030$ | $0.530 \pm 0.038$ |
| EME | $0.530 \pm 0.029$ | $0.982 \pm 0.013$ | $0.620 \pm 0.017$ | $0.928 \pm 0.026$ | $0.610 \pm 0.057$ | $0.760 \pm 0.045$ |
| CTRL-SR | $0.000 \pm 0.000$ | $0.470 \pm 0.039$ | $0.361 \pm 0.070$ | $0.919 \pm 0.038$ | $0.313 \pm 0.010$ | $0.838 \pm 0.033$ |
| TEB (Ours) | $\mathbf{0.931} \pm 0.017$ | $\mathbf{0.982} \pm 0.010$ | $\mathbf{0.984} \pm 0.012$ | $0.968 \pm 0.016$ | $\mathbf{0.878} \pm 0.022$ | $\mathbf{0.917} \pm 0.023$ |

## C.3. Maze2D Experiments

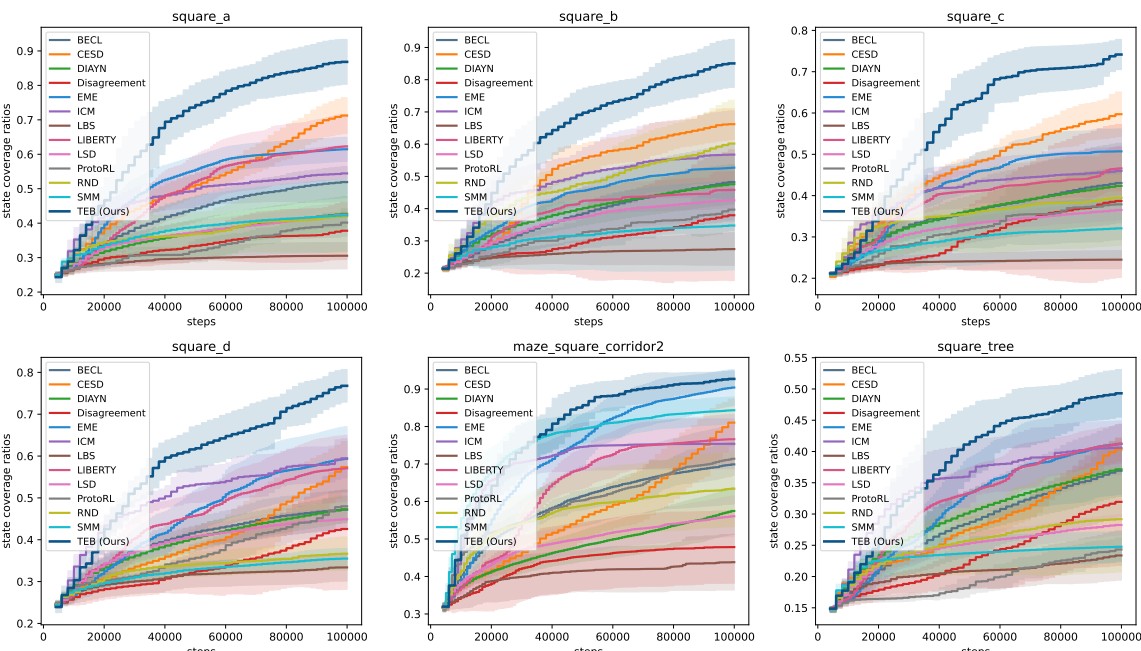

*Figure 7.* Comparison of the learning curves of TEB and baselines in Maze environments with 100k training steps. Each experiment runs with 10 random seeds, with shaded region representing the standard deviation across seeds.

## C.4. Learning Curves in Maze2D

As shown in Figure 7, we also present the learning curves of TEB and the exploration baselines over 100K training steps. We can see that TEB (blue line) significantly outperforms all baselines in both learning speed and maximum exploration coverage across Maze2D tasks, especially in *Square_a* and *Square_c*. Furthermore, we observe that most baselines, such as LBS and RND, converge early and cease generating valuable transitions, while also achieving low exploration capabilities. CeSD, a promising method, does not converge in most tasks, remaining in stable exploration, but its efficiency is relatively low. Overall, the curve comparisons strongly demonstrate that the exploration bonus based on the predictive bisimulation metric yields the strongest exploration abilities.

## C.5. Exploration Visualizations in Maze2D

In Figure 8, we show more visualizations of state coverage in Maze2D tasks.

---

**Algorithm 1 TEB**: Task-Aware Exploration With a Predictive Bisimulation Metric.

---

Initialize encoder $\phi_\omega$ and Replay buffer $\mathcal{D}$, policy $\pi$, transition model $\widehat{\mathcal{P}}$, reward predictor $p_\theta$ and et al.
**for** $m \leftarrow 1$ to (# epochs) **do**
    **for** $i \leftarrow 1$ to (# steps per epoch) **do**
        Encode state $z_t = \phi_\omega(s_t)$
        Execute action $a_t \sim \pi_\phi(z_t) + \epsilon(t)$ where $\epsilon(t) \sim \mathcal{N}(0, \sigma_t^2)$.
        Run a step in environments $s_{t+1} \sim \mathcal{P}(\cdot|s_t, a_t)$
        Collect data $\mathcal{D} \leftarrow \mathcal{D} \cup \{s_t, a_t, r_{t+1}, s_{t+1}, done\}$
    **end for**
    **for** $g \leftarrow 1$ to (# gradient steps per epoch) **do**
        Sample batch $\mathcal{B}_i \sim \mathcal{D}$ and encode $z = \phi_\omega(s)$, $z' = \phi_\omega(s')$
        Update reward predictor by minimizing $\mathcal{L}_{\text{rew}}$ in Eq. (8)
        Update transition model $\widehat{\mathcal{P}}$ by minimizing $\mathcal{L}_{\text{dyn}}$
        Rearrange batch $\mathcal{B}_j = \text{REARRANGE}(\mathcal{B}_i)$
        Compute predictive reward discrepancy $\Delta_R^\pi$ via Eq. (6)
        Compute predictive bisimulation target $y = \text{sg}\left[(\widehat{\mathcal{T}}^\pi \hat{d}_{\bar{\omega}})(s_i, s_j)\right]$ via Eq. (5)
        Update encoder by minimizing $\mathcal{L}_{\text{bisim}}$ in Eq. (10) (and $\mathcal{L}_{\text{rep}}$ if used)
        Compute anchor states (Eq. (13))
        Compute potential $\Phi$ and intrinsic bonus $F = \gamma\Phi(s_{t+1}) - \Phi(s_t)$ (Eq. (12)–(14))
        Update actor-critic using shaped reward $r_t' = r_t + \eta F$
        Update target critics: $Q^{\text{tgt}} \leftarrow \tau Q + (1 - \tau)Q^{\text{tgt}}$ with $\tau$
    **end for**
**end for**

---

## C.6. Baselines Details in Maze2D Experiments

We summarize the background of all exploration and skill discovery baselines used in Maze2D experiments.

**ICM (Pathak et al., 2017)** proposes a curiosity-driven exploration mechanism where an intrinsic reward is defined by the prediction error of a forward dynamics model. This method encourages agents to explore transitions that are hard to predict, reflecting novel experiences.

**RND (Burda et al., 2018)** utilizes a fixed random target network and a trained predictor. The prediction error serves as an intrinsic reward signal, helping agents detect novel states in a scalable and robust manner.

**Disagreement (Pathak et al., 2019)** estimates exploration based on the disagreement among an ensemble of forward models. High disagreement across the ensemble indicates uncertainty, promoting exploration of under-sampled areas.

**ProtoRL (Yarats et al., 2021b)** integrates prototype representations into RL, encouraging exploration toward under-represented clusters in the state space. This structured exploration improves sample efficiency and diversity of learned behaviors.

**LBS (Mazzaglia et al., 2022)** applies Bayesian surprise not in parameter space but in a learned latent space of dynamics. This approach is more computationally efficient and more robust to stochastic environments.

**DIAYN (Eysenbach et al., 2018)** learns diverse skills in the absence of reward by maximizing mutual information (MI) between skills and the resulting states. Each skill should generate distinguishable behavior trajectories .

**SMM (Lee et al., 2019)** aims to flatten the state visitation distribution by minimizing KL divergence between the current state distribution and a target. This entropy-based method improves exploratory coverage.

**LSD (Park et al., 2022)** imposes Lipschitz constraints on skill learning objectives to enforce local continuity. This helps prevent mode collapse and ensures skills generalize across similar states.

**BeCL (Yang et al., 2023)** proposes a contrastive learning objective for skill discovery by treating skill-conditioned trajectories as different views. It maximizes MI between behaviors under the same skill, promoting both diversity and consistency.

*Table 3.* Detailed hyperparameters.

| Shared Hyperparameter | Value in MetaWorld | Value in Maze2D |
|---|---|---|
| Training steps | $1 \sim 3\text{M}$ | 100K |
| Seed frames | 4000 | 4000 |
| Exploration steps | 2000 | 2000 |
| Evaluation episodes | 10 | - |
| Replay buffer capacity | $2 \times 10^5$ | $2 \times 10^5$ |
| Episode length | 1000 | 50 |
| Batch size | 256 | 1024 |
| Frame stack | 3 | 1 |
| Discount factor $\gamma$ | 0.97 | 0.99 |
| State dims | $9 \times 84 \times 84$ | 2 |
| Encoder conv kernels | [32,32,32,32] | - |
| Encoder conv filter size | $[3 \times 3, 3 \times 3, 3 \times 3, 3 \times 3]$ | - |
| Encoder conv strides | [2,1,1,1] | - |
| Hidden dims | 1024 | 1024 |
| Latent state dims | 50 | - |
| Action repeat | 2 | 1 |
| $n$-step returns | 3 | 3 |
| Optimizer | Adam | Adam |
| Learning rate | $1 \times 10^{-4}$ | $1 \times 10^{-4}$ |
| Soft update rate $\tau$ | 0.01 | 0.01 |
| Gradient training frequency | 2 | 2 |
| Linear exploration stddev. schedule | Linear(1.0,0.1,500000) | 0.2 |
| Linear exploration stddev. clip | 0.3 | 0.3 |

**CeSD (Bai et al., 2024)** develops an approach where multiple skills explore distinct clusters of the state space. It introduces distributional constraints to ensure that skills visit non-overlapping regions, thereby improving coverage and distinguishability.

**LIBERTY (Wang et al., 2023)** develops a potential-based exploration bonus using an inverse dynamic differential term, encouraging exploration through state discrepancy.

**EME (Wang et al., 2024a)** proposes a metric-based exploration bonus with a diversity-enhanced scaling factor to improve exploration efficiency and scalability in challenging environments.

## D. Method Details

### D.1. Pseudo-code Algorithm

We describe the TEB algorithm in Algorithm 1.

### D.2. Hyperparameter settings

As shown in Table 3, we provide shared training hyperparameters in MetaWorld and Maze2D. In the Maze experiments, we calculated the state coverage obtained by each algorithm by discretizing the map for each task into tiny bins. Specifically, we set the discrete cell size to 0.1 for *Square_bottleneck* and *Square_tree*, and 0.05 for the others. Additionally, details of TEB-specific hyperparameters are provided below. First, the coefficient $c_R = 1$ is used for the reward differential term, the coefficient $c_T = \gamma$ within the predictive Bisimulation metric. The variance bounds $\sigma_{min} = 1e - 4$ and $\sigma_{max} = 1.0$ are used to train the Gaussian predictor of the reward. The loss weight is $\lambda_r = 0.5$ for the training of Gaussian reward predictor, $\lambda_p = 1e - 4$ for the dynamics model, and $\lambda_b = 0.5$ for the bisimulation metric. Finally, we set the intrinsic reward weight $\eta = 1.0$ for most MetaWorld tasks, except for $\eta = 0.05$ for *Stick-pull* and $\eta = 10$ for *Push-back*. Since the extrinsic reward has been normalized, we set $\eta$ by initial intrinsic reward below 0.05, so the choice of $\eta$ is not complicated.

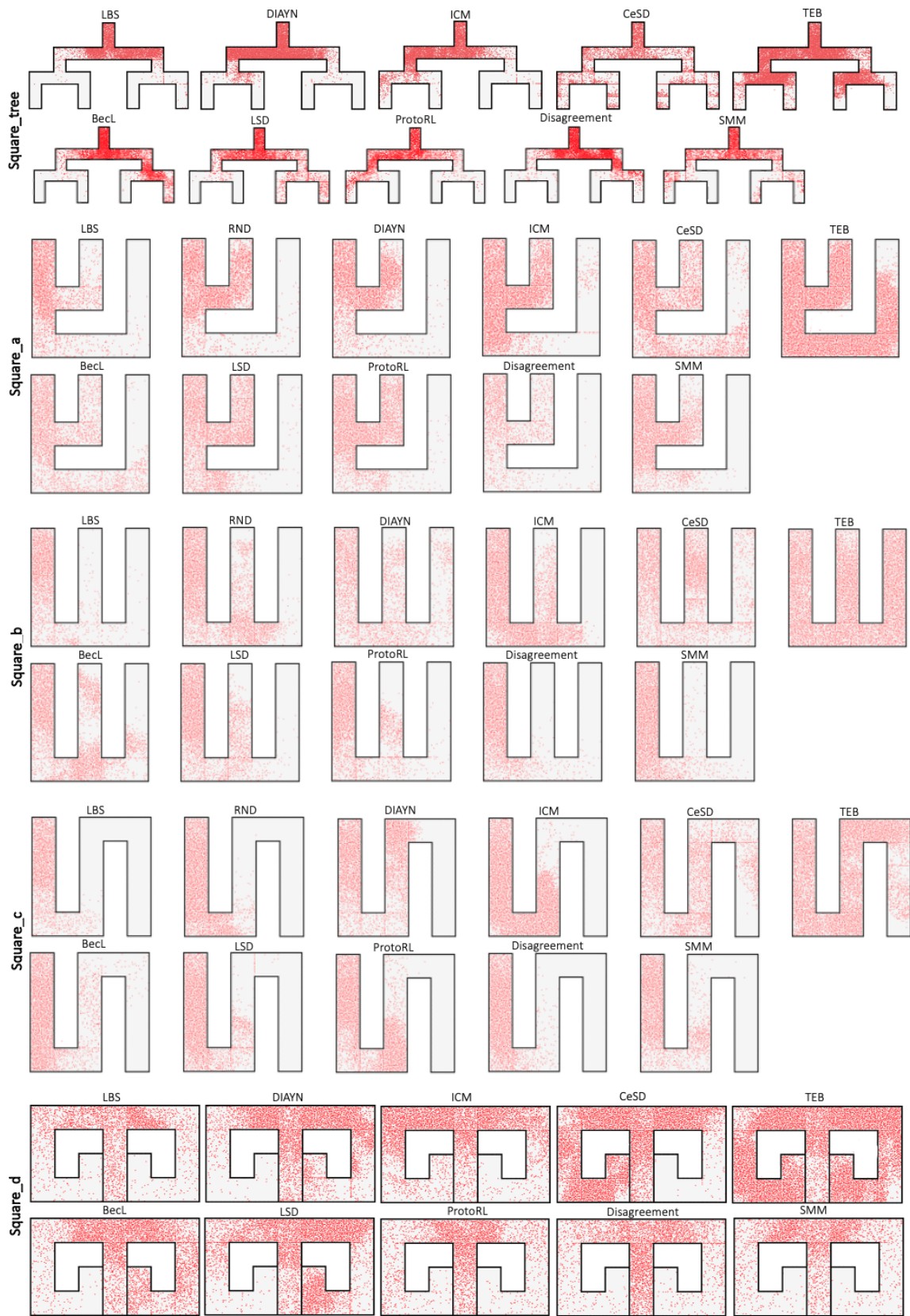

*Figure 8.* Visualization of state coverage in Maze2D tasks.

