# OpenReview forum: "Task-Aware Exploration via a Predictive Bisimulation Metric"
_ICML.cc/2026/Conference — ICML 2026 regular_

### Official Review · Reviewer_gkkn · 2026-02-26

**Soundness:** 2
**Presentation:** 3
**Significance:** 2
**Originality:** 2
**Overall Recommendation:** 4
**Confidence:** 5

**Summary:**

This work focuses on task-aware exploration, especially for environments with sparse reward settings. Existing intrinsic exploration methods often fail in visual RL domains because they either assume access to low-dimensional states or lack strategies that focus specifically on task-relevant information. This work proposes TEB to address this gap by using a predictive Bisimulation metric to tightly couple task-relevant representations with exploration. This work introduces a "predicted reward differential" to theoretically prevent the representation collapse that typically happens to bisimulation metrics under sparse rewards. Extensive experiments on MetaWorld and Maze2D show the effectiveness of TEB.

**Compliance With Llm Reviewing Policy:**

Affirmed.

**Final Justification:**

All my concerns have been addressed and I have raised my score accordingly.

**Key Questions For Authors:**

See weaknesses above. I'd like to adjust my score if the authors can fully addressed my concerns.

**Limitations:**

The authors have discussed the Impact Statement of this work, but I do not find the discussion of the limitation for this work.

**Strengths And Weaknesses:**

Strengths:

- Applying metric-based intrinsic reward is interesting and makes sense for unsupervised exploration.

- Extensive experiments in both sparse reward environments like MetaWorld and pure exploration environments like Maze2d show the effectiveness of TEB.

Weaknesses:

- Lack of some related and important baselines, especially in the experiments of Maze2d. As shown in previous work [1], unsupervised methods for choosing $-log p(s)$ as intrinsic rewards like CIC [2] (apply APT with skill discovery methods) and ExDM [1] (use diffusion models to estimate $-log p(s)$) own larger state coverage in these mazes (for example, in Square-bottleneck, these two methods can achieve 0.58 and 0.75, and this work has cited CIC, but does not compare with CIC in the experiments of Maze2d). Although SOTA performance is not necessary for academic research, it is necessary to discuss these methods and compare with these methods to show the advantages of the proposed method.

- These are several works discussing applying Bisimulation Metric into unsupervised exploration of RL [3-4], but are not included in this work. It will make this work more solid to discuss and compare with these works.

- As downstream task fine-tuning is an important application for unsupervised exploration methods, I'm curious about the performance in these benchmarks (like URLB). (It will not be my major concern of this work)

Reference:

[1] Exploratory Diffusion Model for Unsupervised Reinforcement Learning

[2] Cic: Contrastive intrinsic control for unsupervised skill discovery

[3] Efficient Potential-based Exploration in Reinforcement Learning using Inverse Dynamic Bisimulation Metric

[4] Rethinking exploration in reinforcement learning with effective metric-based exploration bonus

---

> ### Author Rebuttal · Authors · 2026-03-31
>
> Dear Reviewer,
>
> We thank the reviewer for your careful review and your interest in this metric-based intrinsic reward work. We hope that the following statements, explanations, and rich experiments can alleviate your concerns.
>
> > **W1: Lack of some related and important baselines, among other issues.**
>
> **A**: We sincerely thank the reviewer for mentioning the important and strong baselines. We will discuss and compare them in detail in the Appendix. Below, we provide a detailed explanation.
>
> To be honest, **we did not involve these baselines mainly because there are essential differences in the learning objectives between TEB and these methods, while considering that we had the similar baselines (e.g., CeSD) as them.** Specifically, CIC and ExDM aim to maximize state entropy to cover the state space, and thus belong to the task-agnostic exploration. In contrast, TEB focuses on task-aware exploration, and it does not focus on unsupervised skill learning or state coverage itself. **Another reason is the different experimental goals.** The main goal of TEB is to verify the performance in complex dynamic visual environments, such as those with texture, lighting, and shadow distractions. The Maze2D experiment is only to provide an auxiliary verification of the main experiments. Note that when we submitted the initial version, the ExDM codebase had not yet been officially released, so we did not discuss or compare it. Therefore, we selected a comprehensive set of baselines comprising recent visual methods (e.g., CTRL-SR) and unsupervised methods (e.g., CeSD)
>
> > **W2: These are several works discussing applying Bisimulation Metric into unsupervised exploration of RL.**
>
> **A**: Thank you for this valuable guidance. We apologize for not discussing these related works. Below, we provide a brief discussion and thorough performance comparison. In the revision, we will supplement the discussion and comparative experiments to improve our paper.
>
> EME and LIBERTY are both good works, and **we believe that TEB still differs from them in many aspects and has several advantages.** Specifically, TEB aims to address the representation collapse problem of the bisimulation metric under sparse-reward settings without changing the theoretical structure of the bisimulation metric itself, and then builds a task-relevant global explorer on top of it. In contrast, these methods introduce heuristic terms such as a KL term and an inverse-dynamics differential into the bisimulation metric to guide exploration, but they do not address the fundamental issue of representation collapse or the limitation of local exploration.
>
> We strongly agree with the experimental suggestion and are willing to conduct a full comparison with these two baselines. The following two tables report the comparison results (3 seeds) on the MetaWorld and Maze2D, respectively. **Overall, TEB still outperforms these metric-based methods in terms of best episode success rate and maze coverage. Moreover, the convergence speed of TEB is also significantly better than that of EME and LIBERTY in most tasks.** The corresponding curves are shown in Fig. 1 and Fig. 2 in the [`Anonymous Link`](https://anonymous.4open.science/r/ICML_fig-FB7E/README.md). In summary, these results further show that TEB has strong exploration performance.
>
> |Method|Pick-out-of-hole|Pick-place|Push-back|Box-close|Stick-pull|Coffee-push|
> |-|-:|--:|-:|-:|-:|-:|
> |LIBERTY|0.578±0.048|0.860±0.023|0.330±0.009|0.873±0.098|0.824±0.030|0.530±0.038|
> |EME|0.530±0.029|0.982±0.013|0.620±0.017|0.928±0.026|0.610±0.057|0.760±0.045|
> |**TEB**|0.931±0.017|0.982±0.010|0.984±0.012|0.968±0.016|0.898±0.022|0.917±0.023|
>
> |Method|Square-a|Square-b|Square-c|Square-d|Corridor2|Square-tree|Square-bottleneck|
> |-|-:|-:|-:|--:|-:|-:|-:|
> |LIBERTY|0.62±0.16|0.46±0.09|0.47±0.09|0.57±0.13|0.77±0.11|0.41±0.05|0.35±0.04|
> |EME|0.61±0.06|0.53±0.07|0.51±0.09|0.59±0.13|0.90±0.08|0.41±0.13|0.42±0.07|
> |**TEB**|0.87±0.07|0.85±0.07|0.74±0.04|0.77±0.02|0.93±0.02|0.50±0.04|0.47±0.03|
>
> > **W3: Downstream task fine-tuning is an important application for unsupervised exploration methods.**
>
> **A:** Thank you for your suggestion and interest. Here, we briefly supplement the fine-tuning performance of TEB on the URLB benchmark. Following the prior work, we pretrain on the Walker and Quadruped domains with1e5 steps, respectively, and then fine-tune 1e5 and 5e5 steps with 5 seeds on four downstream tasks (e.g., walker_stand, walker_walk, walker_run, walker_flip) in each domain, respectively. **The experimental results show that reward-free training with TEB also can effectively boost the policy performance of the agent during the fine-tuning stage.**
>
> |Method|Walker|Quadruped|
> |-|-:|-:|
> |CeSD|657.3±103.5|766.2±147.9|
> |LIBERTY|645.5±94.1|618.8±102.3|
> |EME|704.8±106.5|673.0±158.4|
> |**TEB**|726.5±109.1|812.3±155.7|
>
> > **Limitations.**
>
> **A**: Thanks for your reminder. We will incorporate the suggestions and discussions to add the limitations of the paper.

---

> > ### Author Rebuttal · Reviewer_gkkn · 2026-04-03
> >
> > I understand that the major contribution of this work is about task-aware exploration, while CIC and ExDM focus on pure exploration. As the Maze2d environment is a pure exploration setting, I still think a detailed comparison with CIC and ExDM is significant, not for achieving SOTA performance, but for better positioning the contribution of this work (the source code of CIC has been available for a long time). I'd like to adjust my score if the authors can better introduce the difference between task-aware exploration like this work and the pure exploration like CIC/ExDM, as well as why the performance of CIC/ExDM is better in Maze2d.

---

> > > ### Author Response · Authors · 2026-04-05
> > >
> > > Dear Reviewer,
> > >
> > > Thank you for your valuable comments, and we agree with your suggestions. In the revision, we will carefully discuss the differences and performance between them, as well as the positioning of TEB. Below, we clarify the fundamental differences between TEB and CIC/ExDM from multiple perspectives, and why CIC/ExDM perform better in Maze2D.
> > >
> > > > **Differences between TEB and CIC/ExDM.**
> > >
> > > **1. Significant differences in learning objectives (primary).**
> > > CIC and ExDM focus on how to maximally discover diverse skills or cover reachable regions, exploring the state space almost uniformly in a task-agnostic manner. In contrast, **TEB is a task-aware exploration method that simultaneously focuses three questions: What is the current abstract task? How to efficiently explore task-relevant regions? And how to better learn task-related policies?** Overall, its objective is to efficiently discover external rewards guided by learned task-relevant information, ultimately improving the learning efficiency of the task policy.
> > >
> > > **2. Differences in applicable scenarios (primary).**
> > > The difference in learning objectives directly leads to different application scenarios. Specifically, TEB is suitable for tasks that require representations under sparse rewards, especially complex visual dynamics, e.g., MetaWorld. In contrast, CIC and ExDM are more suitable for exploration-centric scenarios (e.g., mapping in SLAM), or pure exploration benchmarks such as Maze2D. Additionally, they can serve as pretraining methods, followed by downstream reinforcement learning fine-tuning to accelerate task policy learning in sparse-reward tasks. Overall, evaluating the core performance of these methods requires different experimental scenarios.
> > >
> > > **3. Technical differences.** CIC encourages behavioral diversity by maximizing the MI between skills and states, while ExDM relies on diffusion models to fit diverse replay buffer distributions. Both of these techniques ultimately lead to the indirect maximization of state entropy, based on a task-agnostic manner. In contrast, the key technical distinction of TEB lies in its deep coupling of representation and exploration via the proposed predictive bisimulation metric, while mitigating issues such as both representation collapse and local exploration within the metric, which ultimately serves task-relevant decision performance. Overall, the two approaches have different technical objectives.
> > >
> > > ---
> > >
> > > > **Why do CIC and ExDM perform better in Maze2D?**
> > >
> > > We believe there are two main reasons for this.
> > >
> > > First, in the experiments, **the research objectives of CIC and ExDM naturally align with the evaluation metrics, i.e., state coverage of the Maze2D benchmark**. As discussed above, CIC and ExDM aim to cover the state space as much as possible via ingenious and efficient modeling. In contrast, TEB is designed for sparse-reward settings with visual representations, e.g., MetaWorld, as they can fully reflect the contribution of the predictive bisimulation metric to policy performance in terms of representation and exploration.
> > >
> > > In the table below, we provide a simple comparison between TEB and CIC on the visual MetaWorld environment with sparse rewards (3 seeds, 1M steps). **The results of task success rates provide preliminary evidence that TEB outperforms the CIC method that is based on a task-agnostic exploration paradigm in the complex visual tasks.** This is intuitive because arbitrarily exploring the entire action space of each arm joints by a task-agnostic manner would actually lead to inefficiency even a dead end.
> > >
> > >
> > > | Method | Box-close       | Pick-out-of-hole |
> > > |--------|-----------------|------------------|
> > > | CIC    | 0.631 ± 0.025   | 0.584 ± 0.049    |
> > > | **TEB**    | 0.968 ± 0.016   | 0.931 ± 0.017    |
> > > |
> > >
> > > Second, in the Maze2D environment, **the representation ability of the predictive bisimulation metric in TEB is not activated**. Instead, only its metric-based global exploration component is utilized, which does not fully reflect the systemic strengths of TEB. As noted earlier, the role of Maze2D in our experiments is mainly supplementary, aiming to demonstrate the effectiveness of the pseudo-anchor based global intrinsic exploration reward in TEB.
> > >
> > >
> > > We sincerely hope you will carefully consider the significant differences between TEB and CIC/ExDM. Thank you very much.

---

### Official Review · Reviewer_qSSs · 2026-03-12

**Soundness:** 3
**Presentation:** 3
**Significance:** 3
**Originality:** 4
**Overall Recommendation:** 5
**Confidence:** 4

**Summary:**

The general idea is to learn a representation that abstracts task-irrelevant variation in state features, so that an exploration bonus does not reward task-irrelevant novelty and instead prioritizes meaningful behavioral differences. To do so, the paper proposes a predictive bisimulation metric called TEB to quantify intrinsic notions of novelty and use it as a potential-based exploration bonus, which has the added benefit of mitigating representational collapse in sparse-reward environments. TEB essentially consists of learning a reward model $r(s)$ and using the difference in predicted rewards instead of the ground truth rewards in the bisimulation metric. Rather than a deterministic prediction, they use distributional reward modeling with Gaussian distributions with learnable mean and variance — they show that this stochasticity prevents collapse in sparse reward environments by injecting an “energy floor” that keeps the reward differential term from being 0 everywhere early on.

To show the effectiveness of this exploration bonus, they demonstrate fast convergence and higher success rates in MetaWorld, and more state space coverage in reward-free Maze2D tasks.

**Compliance With Llm Reviewing Policy:**

Affirmed.

**Final Justification:**

My main concerns about this paper were writing clarity and lack of clarity on some theoretical justifications and intuitions for preventing collapse. I found the authors' answers satisfactory, and, given the promised changes, am happy to maintain my score as an accept.

**Key Questions For Authors:**

1. Can you expand on why the metric alone might encourage only local transitions (right column, line 211)? Is it because the standard metric only measures the difference between $s$ and $s'$ separated by a single timestep, whereas the intra-batch method computes global novelty compared to a batch of randomly sampled states in the environment that may be an arbitrary distance apart?
2. Can you justify the preservation of the optimal policy under the per-batch novelty score? A standard bonus that is a function only of the state $\Phi(s_t)$ does preserve this and follows directly from Ng et al. (1999), but I am unsure whether or not this holds for a per-batch “baseline.”
3. How does TEB, or the bisimulation metric more broadly, work in the reward-free setting? How are the reward predictors fit? Is it purely relying on the stochasticity of the Gaussian that is always centered at 0 or some other fixed value to produce a reward differential?
4. Can you explain the rationale for leaving out other strong metric learning-based exploration algorithms such as METRA [1] and CSF [2] from the comparisons in Figure 3 and Table 1? METRA in particular might be a more similar baseline for state coverage purposes since it directly learns a metric for exploration.

**References**

1. Park et al. (2024). METRA: Scalable Unsupervised RL with Metric-Aware Abstraction. ICLR 2024.
2. Zheng et al. (2025). Can a MISL Fly? Analysis and Ingredients for Mutual Information Skill Learning. ICLR 2025.

**Limitations:**

Yes

**Strengths And Weaknesses:**

**Strength**

The idea of using a form of “group relative” novelty score is quite nifty and interesting in my opinion, and an interesting way to estimate the novelty and encourage more global exploration. The empirical results are strong and demonstrate fast convergence relative to baseline algorithms. Overall, I think this is a good paper with significant novel contributions.

**Weaknesses**

The reward differential should be defined when it is first mentioned. While it can be inferred easily from Equation 5 onwards, it would be good to explicitly define it earlier.

Additionally, I have some theoretical concerns: while a fixed reward potential that is a function of the state is guaranteed to preserve policy optimality, is this true when the bonus is relative to a per-batch average, which seems to add a dependence on the stationary visitation distribution?

Reproducibility: three random seeds seems quite low given the spread of the standard deviations. While the results do look promising, I think it would be good to run additional seeds to ensure significance.

---

> ### Author Rebuttal · Authors · 2026-03-31
>
> Dear Reviewer,
>
> Thank you for the appreciation of the idea of using a global pseudo-state anchor to measure the “group relative” novelty score, and thank you for your positive evaluation of the experimental results. We respond point by point to your other concerns and specific questions as follows.
>
> > **W1: The reward differential should be defined when it is first introduced.**
>
> **A**: Thank you for this helpful suggestion. We fully agree. In the revision, we will introduce the definition of the reward differential when the concept is first mentioned.
>
> > **W3: Discussion of experimental seeds.**
>
> **A**: We understand this concern. We hope the following clarification can alleviate it. In the MetaWorld, **although the paper reports results based on 3 seeds, in practice we ran TEB and key baselines, such as RAP and DrM, for more than 3 seeds.** The purpose is that we also want to more accurately verify the effectiveness of TEB and its improvement over the baselines through repeated runs. Notably, in Maze2D, where we evaluate reward-free pure exploration, the reported results are averaged over 10 seeds, which provides a much more reliable indication of the superior exploration performance of TEB relative to the baselines. Nevertheless, in the revision, we will discuss the limitation related to random seeds.
>
> > **Q1: Can you expand on why the metric alone might encourage only local transitions?**
>
> **A**: Thank you for this comment. Your understanding is very accurate. If one only uses $d(s_t, s_{t+1})$ as the intrinsic exploration reward, it essentially encourages the agent to seek state pairs with large task-relevant changes. As a result, **the agent may get trapped between the two states and obtain intrinsic rewards by frequently transitioning between them**, while such rewards do not substantially help discover new regions. TEB changes this local limitation by introducing the batch-based pseudo-state anchor $s_*$. In the revision, we will add this explanation to the corresponding section.
>
> > **Q2: Can you justify the preservation of the optimal policy under the per-batch novelty score?**
>
> **A**: Thank you for this very insightful theoretical observation. **We believe that the policy invariance can still be supported under a weak static assumption together with the theory of dynamic potential functions [1].** An informal proof sketch is as follows. First, at each batch training step of policy optimization, we treat the anchor $s_\star$ as a fixed global reference. In this case, $\Phi(s; s_\star)$ is a standard potential function that depends only on the state within the current update period. Second, according to the extension of the theory of Ng et al. by Devlin et al. (2012) [1], even if the potential function changes over time, i.e., $s_\star$ evolves with the training distribution, policy invariance can still be preserved as long as $s_\star$ eventually becomes stable or converges, because its influence on the optimal policy vanishes. In TEB, as the representation learning and the policy gradually converge, the batch mean $s_\star$ also tends to become stable, which supports the above informal argument.
>
> > **Q3: How does TEB work in reward-free settings?**
>
> **A**: We apologize for not explaining this clearly enough. In the revised version, we will add a more explicit description of how the reward predictor in TEB works under the reward-free setting. Specifically, in reward-free tasks, the Gaussian reward predictor still uses the sparse external reward as its reward target, and note that the reward is in practice almost zero everywhere. Therefore, it nearly relies on a constant value, and the reward differential is mainly produced by the stochasticity of the Gaussian distribution. A more detailed understanding can be gained by combining Theorem 3.3 with the associated analysis.
>
> > **Q4: Can you explain the rationale for leaving out other strong metric learning-based exploration algorithms?**
>
> **A**: Thank you for this meaningful question. We hope the following explanation can alleviate your concern. Our consideration was as follows. To be frank, CeSD, METRA, and many similar baselines all belong to the intrinsic exploration paradigm based on maximizing state entropy. We therefore chose the more recent CeSD baseline and did not further include more baselines of the same type, although we acknowledge the existence of many strong, similar baselines. In addition, since TEB focuses more on intrinsic exploration under complex visual observations, among recent methods we selected CTRL-SR, which is also in the representation learning line and aims to alleviate representation and exploration difficulties through spectral representations. Overall, we aimed to keep the selected baselines diverse and recent under a reasonable comparison setting.
>
>
> **Reference**
>
> 1. Sam Devlin and Daniel Kudenko. Dynamic potential-based reward shaping. In AAMAS 2012.

---

> > ### Author Rebuttal · Reviewer_qSSs · 2026-04-04
> >
> > I appreciate the clarifications and my concerns about seeds, local transitions, reward-free settings have been resolved. I find the authors' explanation of the the potential-based optimality preservation reasonable combined with the convergence guarantees of bisimulation metrics, and it would be good to have a more formal discussion of this in the paper. I do think that maximization of state entropy is an oversimplification/incorrect characterization of previous methods such as METRA, which explicitly maximize temporal distances and therefore a meaningful notion of state coverage. While I acknowledge that TEB tackles a different problem (task-related exploration), it would still be good to discuss these distinctions.

---

> > > ### Author Response · Authors · 2026-04-04
> > >
> > > Dear Reviewer,
> > >
> > > We sincerely thank you for the detailed feedback and for acknowledging that our clarifications have resolved most of your concerns.
> > >
> > > - We appreciate your valuable suggestion regarding optimality preservation. In the revised version, we will supplement a more formal and comprehensive theoretical analysis and discussion regarding optimality preservation.
> > >
> > > - We agree with your point, particularly that METRA should not be simply categorized as a state-entropy maximization method. METRA is designed for unsupervised, task-agnostic exploration, where it maximizes temporal distances between skill trajectories to induce a meaningful notion of state coverage in a compact latent space, without relying on task or reward signals. In contrast, TEB focuses on task-aware exploration under sparse rewards in visual RL settings. As a result, TEB is particularly suited for environments such as MetaWorld, where substantial visual distractions (e.g., lighting, textures) and sparse rewards are present. We agree that discussing these distinctions is meaningful, and we will provide a more rigorous analysis of METRA and its distinctions from TEB in the revised version.

---

### Official Review · Reviewer_pkgw · 2026-03-13

**Soundness:** 3
**Presentation:** 3
**Significance:** 2
**Originality:** 2
**Overall Recommendation:** 5
**Confidence:** 3

**Summary:**

The authors introduce Task-aware Exploration through a predictive Bi-simulation (TEB) metric. TEB intrinsically rewards reaching states that maximize the bisimulation metric to the anchor state, a metric that collapses/combines representations of states with similar rewards and dynamics (e.g. states with similar tasks). The authors benchmark TEB relative to several exploration algorithms based on intrinsic reward maximization in Metaworld and Maze 2D and shows TEB has state-of-the-art success rates when paired with standard rewards and state coverage. There are also some ablation experiments testing various design choices.

**Compliance With Llm Reviewing Policy:**

Affirmed.

**Final Justification:**

The rebuttal answered some key confusions I had about the paper, specifically the methods presentation, and helped clarify significance relative to other methods. They showed clear evidence on why, conceptually, the bi-simulation is useful for generalization with didactic experiments (e.g. in settings like the T-shaped maze).

The clarifications and the evaluation of generalization raised by score from 3->5 and I support acceptance.

**Key Questions For Authors:**

See questions in Weaknesses. I mostly have questions about the actual formulation of the objective, and the meaning + interpretation of results and ablations. I also have some design decision questions.

**Limitations:**

yes

**Strengths And Weaknesses:**

Strengths:

- Intrinsic reward for exploration based on maximizing the bisimulation metric has, to my knowledge, not been done before. (If this is true or false, it is worth it to make this contribution clearer early in the paper). Given desirable properties of representations trained using the bisimulation metric in prior works, this seems like an interesting approach.

- Thorough theoretical results supporting the convergence and boundedness of the policy trained to maximize TEB and TEB.

- Evaluation over Metaworld benchmarks.

Weaknesses:

The main idea and implementation of the exploration method gets lost within the presented technical details in Section 3.

- It is unclear to me where the representation alignment (3) shows up in the final loss (shortly after (10)). Is J_{bisim} simple added to \mathcal{L}_{rep}? This similarly makes the ablations in Fig. 4 confusing -- I'm not really sure what is being ablated, and what counts as the bisimulation metric-based representation learning component and the metric-based intrinsic reward. Though the empirical results show that having both representation alignment and metric based intrinsic reward is important, the argument/reasoning for why both reward components are necessary is buried.

- Why does TEB leads to superior maze coverage in Maze2D tasks? Although it seems beneficial to do exploration with respect to a potential that incentivizes reaching states far (via rewards + transitions) from an anchor state, why is bisimulation a good choice over metrics defined over the state spac like, e.g., the temporal distance quasimetric in "Episodic Novelty Through Temporal Distance" by Jiang et al.? Are there simple didactic examples where other metric-aware exploration like ETD would fail, but TEB would succeed? Or nice generalization properties of bisimulations (e.g. Hansen-Estruch et al. "Bisimulation Makes Analogies in Goal-Conditioned Reinforcement Learning") that would transfer to an exploration setting? Figure 1 is also a bit confusing and did not help me gain intuition on the answer to these questions.

- Why the potential shaping formulation for the intrinsic reward? This makes sense to me when simultaneously maximizing the intrinsic reward and some desired extrinsic reward to keep the optimal policy invariant, but makes less sense to me in the reward-free exploration. Why not directly maximize d_pre(s_{t+1}, s_{anchor})?

**If some of these Weaknesses/questions are addressed to improve the presentation and significance of the work, I am willing to raise my score.**

---

> ### Author Rebuttal · Authors · 2026-03-30
>
> Dear Reviewer,
>
> Thank you for recognizing the novelty, the theoretical completeness and the experiments of this work. We hereby provide a point-by-point clarification to address your concerns.
>
> > **Strength1: Contribution of maximizing the bisimulation metric in our TEB.**
>
> **A**: Thank you for this reminder. In the revised version, we will further clarify the unique contribution of TEB. We hope to further strengthen the our contribution by the following aspects.
>
> **a) Preserving the theoretical purity of the standard bisimulation form.** Strictly speaking, we are the first to use the standard bisimulation metric form (composed of a reward differential and a dynamic differential) for task-aware exploration, although there have been recent related discussions and exploration works, such as EME and LIBERTY mentioned by Reviewer gkkn. Unfortunately, **they often introduce heuristic terms such as a KL term and an inverse-dynamics differential into the bisimulation metric to achieve better performance. They do not follow the standard form and may break its operator properties.**
>
> **b) Mitigating the metric failure and the trap of local exploration.**
> In contrast, we reveal that the **predictive reward differential $\Delta_R$ can mathematically alleviates the typical problems of metric failure** and representation collapse under sparse rewards, whereas EME and LIBERTY do not address it. In addition, directly introducing a metric-based intrinsic reward in the above works can easily make the agent short-sighted. To address this, we **propose a pseudo-state anchor to measure the relative global novelty**, preventing the agent from falling into local exploration.
>
> > **W1: Regarding the explanation of the loss composition and the ablation study.**
>
> **A:** We sincerely apologize for the confusion caused by our writing, and we will carefully polish this part in the revision. Below, we provide a clear explanation of your questions and concerns.
>
> **a) Explanation of the objective loss.** The loss $L_{bisim}$ (Eq. 10) is an equal and specific form of the general objective $J_{bisim}$, which is directly added to the loss $L_{rep}$. We will unify $J_{bisim}$ with $L_{bisim}$ in the revision.
>
> **b) Explanation of the ablation structure.** In Fig. 4, our metric-based representation learning component "Rep" refer to the loss $L_{rep}$. For example, "TEB w/o Rep" represents "TEB, but excluding the metric-based representation loss $L_{rep}$.". The "Int" refers to the intrinsic reward in Eq. (14).
>
> **c) Why both components are necessary?** In complex visual tasks, if the latent space is not aligned by the bisimulation metric ("Rep"), it will make exploration inefficient due to task-irrelevant visual noises. In addition, even with a clean latent space, without an effectively global intrinsic reward ("Int"), the DRL still remains inefficient.
>
> > **W2: Why does TEB leads to superior coverage? And other issues.**
>
> **A:** We thank the reviewer for the insightful comments.
>
> **a) TEB versus Temporal Distance.**
> ETD is an excellent method, but it is essentially task-agnostic. In Maze2D, some regions may be temporally far away, such as dead ends deep in the maze, but still have similar reward potential or dynamics to the current region. TEB compresses such regions by the bisimulation metric, allowing the agent to focus more on regions with higher state value. In contrast, ETD may still be attracted to temporally distant but task-irrelevant regions.
>
> **b) Interpretation from the perspective of generalization.**
> Yes, we agree with this interpretation. The agent can recognize that some regions have already been explored at a functional level, which is difficult to achieve with purely state-space distance measures such as ETD.
>
> **c) Clarification of Fig. 1.**
> The left part of Figure 1 shows a representation space where noise has not been compressed by the bisimulation metric. The right part shows that, after such compression, the task representation space becomes more suitable for computing a task-relevant global intrinsic reward.
>
> > **W3: Why does the intrinsic reward need to adopt the potential-based shaping?**
>
> **A**: Thank you for the meaningful question. **Potential-based shaping serves not only to ensure policy invariance but also plays an active role in intrinsic exploration**, and it is also reasonable in reward-free exploration. Specifically, if we directly reward $d_{pre}(s, s_{anchor})$, the agent tends to seek one or a few points that are farthest from the anchor and may stay there, as staying there can also continuously obtain intrinsic rewards (as the distribution of $s_{anchor}$ changes slowly). In contrast, under the potential-based difference form, the agent receives a reward only when the current state is different from a random state in terms of the distance to the global anchor (avoiding homogenization). This forces the agent to keep moving toward other unknown boundaries, rather than staying at a known distant state.

---

> > ### Author Rebuttal · Reviewer_pkgw · 2026-04-01
> >
> > Thank you for the very thorough response. I have raised my score to a 4, given that these revisions for clarity will be added to the paper for readability. The other reviewer discussions also helped me better appreciate the reward differential idea, and I am generally curious about how this type of approach can help mitigate representation collapse for arbitrary metric-based exploration methods.
> >
> > While I'm not familiar with the EME and LIBERTY methods mentioned by gknn, the author response has sufficiently addressed this point for me. A thorough discussion of this in the paper, and perhaps moving some theoretical details into the appendix, would make the work more compelling.
> >
> > My remaining ask is a didactic experiment to really show **Interpretation from the perspective of generalization**, which I think is important. E.g., can you demonstrate that exploration in setting A automatically leads to performance transfer to setting B following the analogies intuition? **Experiments of this flavor, that highlight the benefits of using the bisimulation metric in particular, would lift me from a 4 to a 5**, though I support acceptance regardless.

---

> > > ### Author Response · Authors · 2026-04-06
> > >
> > > Dear Reviewer,
> > >
> > > We are pleased to have resolved most of your concerns and thank you for your positive feedback. We will add a detailed discussion and experimental comparison results regarding the EME and LIBERTY methods to the paper and appendices to make the paper more convincing.
> > >
> > > We also sincerely thank you for your insightful comments regarding both the fundamental generalization ability of TEB and its higher-level generalizable exploration ability. We meticulously design two sets of experiments to systematically demonstrate that the generalization and generalizable exploration abilities in setting A automatically lead to performance transfer to setting B following the analogies intuition.
> > >
> > > >**1. First, we aim to verify that our bisimulation metric-based TEB provides strong generalization ability, enabling analogies across behaviorally/functionally similar scenarios.**
> > >
> > > To facilitate scenario settings, we conduct experiments on the visual Distracting DMC environment. Specifically, we train TEB using two distracting video backgrounds (as scenario A), and evaluate it on 30 unseen distracting backgrounds (as scenario B). We also compare against a non-bisimulation baseline (DrQ-v2 backbone) under the same setting. All experimental results were run 3 seeds and trained 500k frames.
> > >
> > > As shown in the table below, we report the best episode reward on both the training scenario A and the unseen scenario B over two tasks. We observe that when transferring to unseen scenario B, TEB maintains strong performance with less than 10% degradation. In contrast, the baseline not only struggles to learn a good policy even in scenario A, but also suffers significant performance drops when transferred to scenario B. **These results quantitatively demonstrate that TEB can generate strong scenario generalization abilities, thus performing well on unseen backgrounds.**
> > >
> > > | Method |  | walker_walk |  | | reacher_easy  |  |
> > > |--------|-------------|--|--|--------------|--|--|
> > > |        | Scenario A  | Scenario B | (AtoB) Decay Ratio | Scenario A | Scenario B | (AtoB) Decay Ratio |
> > > | DrQ-v2 | 472.0 ± 38.4 | 279.3 ± 37.1 | ↓40.83% | 581.4 ± 53.3 | 474.3 ± 57.9 | ↓18.42% |
> > > | **TEB**    | 876.9 ± 41.3 | 793.5 ± 35.0 | ↓9.85%  | 940.7 ± 46.1 | 893.5 ± 45.0 | ↓5.02%  |
> > > |
> > >
> > >
> > > In addition, we provide qualitative visualization of the functional/behavioral analogy abilities learned by the encoders of TEB and the non-bisimulation method from the generalization perspective across the two scenarios. As shown in the Figure 3  ([`Anonymous Link`](https://anonymous.4open.science/r/ICML_fig-FB7E/README.md)), we use t-SNE to visualize the latent embeddings of the same batch of observations (1000 samples from scenarios A/B) in the walker_walk task. We further illustrate several pairs of samples from different scenarios but similar in behavior/function. We observe that bisimulation-based TEB (left) maps these sample pairs to nearby positions in its latent space, while the non-bisimulation method does not show this clearly. **This qualitatively indicates that TEB learns functional/behavioral analogies that are invariant to visual disturbances or scenarios.** Note that the color of each embedding point in the figure represents the state value (lighter colors indicate higher value), which we not discuss in detail as it is not the focus.
> > >
> > > >**2. Second, we further verify that TEB’s generalizable exploration abilities can automatically lead to performance transfer across scenarios.**
> > >
> > > We again use the Distracting DMC with scenarios A and B settings. Specifically, we first pretrain TEB and a non-bisimulation unsupervised exploration baseline (CeSD) for 300k frames in scenario A. Then, using frozen encoders trained in scenario A , we fine-tune a same DrQ-v2 for another 300k frames in scenario B, and then report the best reward after fine-tuning.
> > >
> > > As shown in the table below, we observe that the encoder learned by TEB significantly improves downstream policy performance compared to CeSD. **Intuitively, the improvement suggests that TEB is able to transfer both task-relevant representations and generalizable exploration abilities learned in scenario A to scenario B, thereby achieving better DrQ-v2 performance even under the challenging setting where the encoder is frozen.**
> > >
> > > | Method | walker_walk        | reacher_easy       |
> > > |--------|--------------------|--------------------|
> > > | CeSD   | 308.6 ± 56.8       | 427.6 ± 47.3       |
> > > | **TEB**    | 575.9 ± 51.0 (↑86.6%) | 764.4 ± 35.9 (↑78.8%) |
> > > |
> > >
> > > Through the two sets of experiments and visualizations above, we believe that we can effectively verify the behavioral/functional analogy ability of TEB across different scenarios from a generalization perspective. Finally, I hope this will fully resolve your concerns.

---

### Decision · Program_Chairs · 2026-04-30

**Decision:**

Accept (regular)

**Comment:**

This paper is technically sound, well-written, novel, and interesting to some fraction of the ICML community.